# Transcription factor MITF and remodeller BRG1 define chromatin organisation at regulatory elements in melanoma cells

Patrick Laurette[1†], Thomas Strub[1,2†], Dana Koludrovic[1], Céline Keime[1], Stéphanie Le Gras[1], Hannah Seberg[3], Eric Van Otterloo[3], Hana Imrichova[4], Robert Siddaway[5], Stein Aerts[4], Robert A Cornell[3], Gabrielle Mengus[1], Irwin Davidson[1]*

[1]Department of Functional Genomics and Cancer, Institut de Génétique et de Biologie Moléculaire et Cellulaire, Strasbourg, France; [2]Department of Oncological Sciences, Icahn School of Medicine at Mount Sinai, New York, United States; [3]University of Iowa College of Medicine, Iowa City, United States; [4]Laboratory of Computational Biology, Center for Human Genetics, University of Leuven, Leuven, Belgium; [5]Arthur and Sonia Labatt Brain Tumor Research Centre, Peter Gilgan Centre for Research and Learning, Hospital for Sick Children, Toronto, Canada

*For correspondence: irwin@igbmc.fr

†These authors contributed equally to this work

**Abstract** Microphthalmia-associated transcription factor (MITF) is the master regulator of the melanocyte lineage. To understand how MITF regulates transcription, we used tandem affinity purification and mass spectrometry to define a comprehensive MITF interactome identifying novel cofactors involved in transcription, DNA replication and repair, and chromatin organisation. We show that MITF interacts with a PBAF chromatin remodelling complex comprising BRG1 and CHD7. BRG1 is essential for melanoma cell proliferation in vitro and for normal melanocyte development in vivo. MITF and SOX10 actively recruit BRG1 to a set of MITF-associated regulatory elements (MAREs) at active enhancers. Combinations of MITF, SOX10, TFAP2A, and YY1 bind between two BRG1-occupied nucleosomes thus defining both a signature of transcription factors essential for the melanocyte lineage and a specific chromatin organisation of the regulatory elements they occupy. BRG1 also regulates the dynamics of MITF genomic occupancy. MITF-BRG1 interplay thus plays an essential role in transcription regulation in melanoma.

## Introduction

Microphthalmia-associated transcription factor (MITF), a basic helix-loop-helix leucine zipper (bHLH-Zip) factor, regulates specification, survival, and proliferation of normal melanocytes, and controls proliferation, migration and invasion of melanoma cells (*Goding, 2000*; *Widlund and Fisher, 2003*; *Steingrimsson et al., 2004*). The level of functional MITF expression determines many of the proliferation and invasion properties of melanoma cells (*Hoek and Goding, 2010*) and siRNA-mediated MITF silencing induces senescence in several melanoma lines (*Strub et al., 2011*).

We previously reported a genome wide analysis of MITF target genes in 501Mel melanoma cells (*Strub et al., 2011*). Chromatin immunoprecipitation coupled to deep sequencing (ChIP-seq) identified MITF binding sites and integration with RNA-seq following siRNA-mediated MITF knockdown showed that MITF directly and positively regulates genes involved in DNA replication and repair and mitosis. In contrast, MITF represses genes involved in melanoma invasion.

**eLife digest** Melanocytes are pigment-producing cells found primarily in the skin. Many of the genes that help these cells to develop are also thought to affect the development of melanomas: an aggressive form of skin cancer that originates in these cells. One such gene encodes a protein called MITF. This protein binds to DNA and regulates genes that control the development, survival, and spread of melanocytes; it is also linked to the invasive properties of melanomas.

The MITF protein works together with partner proteins to control numerous genes, activating some while inhibiting others, by binding to nearby stretches of DNA that act as regulatory elements. Its interactions are therefore widespread and complex. Now, Laurette, Strub et al. have used techniques called tandem affinity purification and mass spectrometry to identify the proteins that interact with MITF. This investigation found many new protein partners for MITF, including proteins involved in DNA damage, repair, and replication. MITF also associates with two proteins—one of which is called BRG1—that are involved in modifying how tightly DNA is packaged inside cells. DNA wrapped around proteins is known as chromatin, and if chromatin is tightly packed, the genes in that stretch of DNA cannot be easily accessed or activated.

Removing BRG1 from melanocytes and melanoma cells caused the cells to die or stop growing. When BRG1 was removed from developing mouse embryos, melanocytes failed to form. Further investigation revealed that MITF, together with another protein, localize BRG1 to sites in the melanocyte's DNA to open up the chromatin and regulate nearby genes. Furthermore, Laurette, Strub et al. report that BRG1 binds to many such elements in a characteristic manner, in which two BRG1 proteins flank the stretch of DNA bound by MITF and several other key DNA-binding proteins that together regulate many aspects of melanocyte and melanoma cell physiology.

Laurette, Strub et al. have therefore revealed many details about the molecules that activate genes in melanomas and melanocyte cells, as well as the interactions between these molecules. The results could also help researchers to understand how the BRG1 protein organises chromatin packing in other cell types.

To better understand the molecular function of MITF, we used tandem affinity purification to isolate MITF from 501Mel melanoma cells and mass spectrometry to identify its interacting partners. We report here the first comprehensive characterisation of the MITF interactome and we show that MITF interacts physically and functionally with a novel form of the PBAF chromatin-remodelling complex specific for neural crest derived cells comprising both BRG1 and CHD7. We also show that MITF and SOX10 recruit BRG1/PBAF to the nucleosomes flanking critical enhancer elements in the melanocyte lineage.

## Results

### MITF associates with multiple complexes involved in transcription, DNA replication and repair

Using 501Mel melanoma cell lines stably expressing an N-terminal FLAG-HA epitope-tagged MITF (F-H-MITF) to levels comparable to that of endogenous MITF (*Figure 1A*, line J), we performed tandem affinity purification of soluble nuclear-and chromatin-associated fractions (*Drané et al., 2010*). Mass spectrometry and silver nitrate staining showed numerous proteins in the tandem immunoprecipitation from F-H-MITF cells, while almost no proteins were detected in immunoprecipitations from untagged 501Mel cells (*Figure 1B* and *Supplementary file 1*). Two independent purifications and mass spectrometry analyses were performed and only proteins identified specifically in the F-H-MITF precipitations with no peptides in the control precipitations are discussed. In addition, contaminating ribo-nucleoprotein particles, spliceosome components, and chaperone proteins have been excluded from the subsequent analysis.

Previously described MITF partners such as its heterodimerisation partners TFEB, TFE3 and TFEC (*Steingrimsson et al., 2002*) and its cofactor β-catenin (CTNNB1) (*Schepsky et al., 2006*) (*Figure 1D* and *Supplementary file 1*) were detected. We also identified novel potential MITF partners. The BPTF, SMARCA1 (SNF2L), SMARCA5 (SNF2H), and RBBP4 components of the NURF chromatin-remodelling

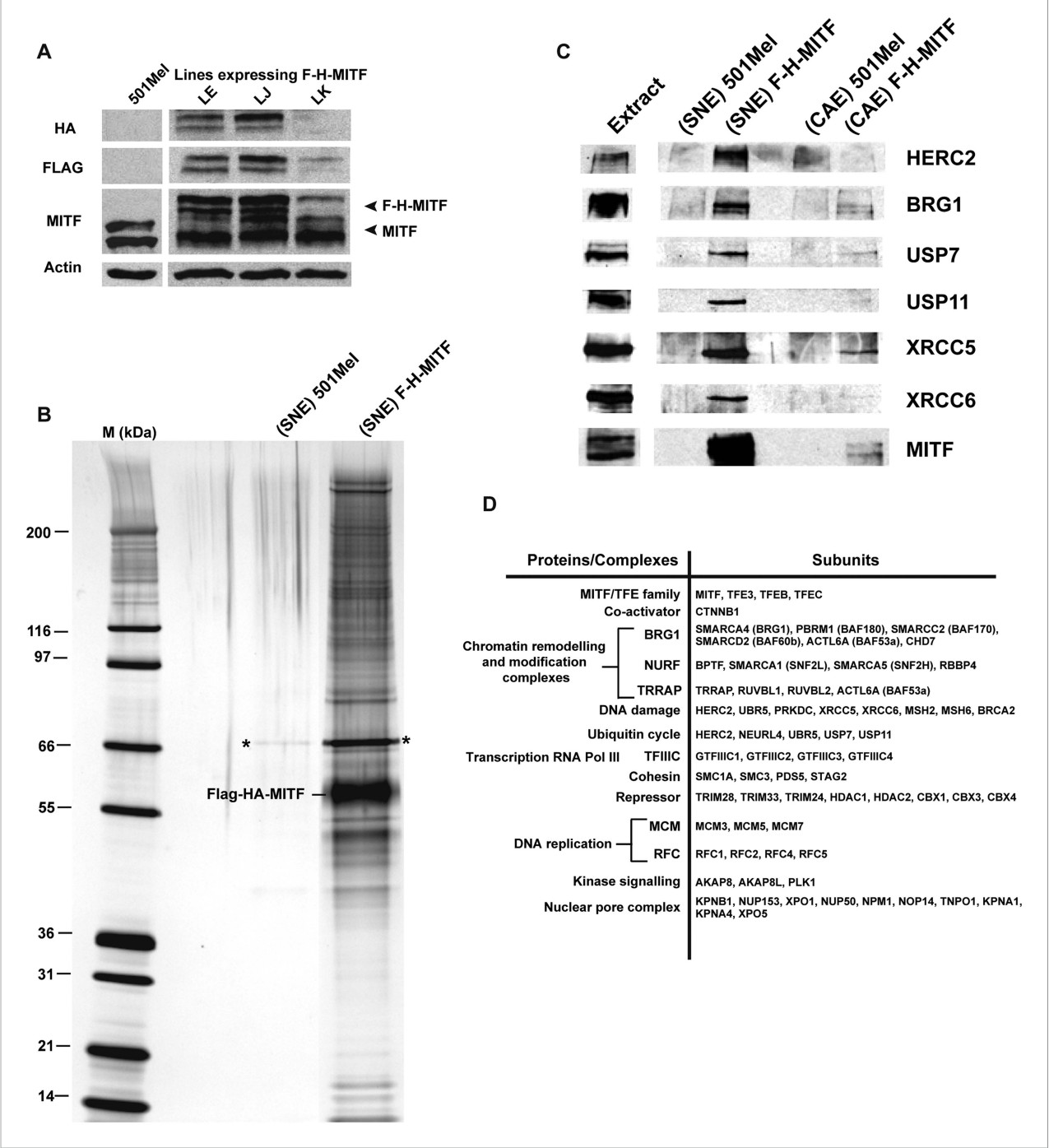

**Figure 1**. Purification of MITF-associated complexes. (**A**) Western blot of 501Mel cell lines stably expressing Flag-HA-tagged-MITF (F-H-MITF). (**B**) The immunoprecipitated material from the soluble nuclear extract (SNE) was separated by SDS PAGE and stained with silver nitrate. F-H-MITF is indicated along with * that designates a contaminating protein seen in the control immunoprecipitations. Lane M corresponds to a molecular mass marker indicated in kDa. (**C**) Immunoblot detection of HERC2, BRG1, USP7, USP11, XRCC5, and XRCC6 in the MITF-associated complexes. (**D**) Summary of proteins and complexes interacting with MITF. Shown are the proteins found specifically in the immunopurifications of F-H-MITF classified according to their function and organisation into known complexes.

complex associated with MITF specifically in the chromatin fraction. Components of the DNA damage response machinery including XRCC5 and XRCC6 (Ku80 and Ku70), DNA-dependent protein kinase (PRKDC), BRCA2 as well as MSH2 and MSH6 were identified along with the HECT domain-containing

E3-ligase HERC2, previously implicated in DNA repair (*Bekker-Jensen et al., 2010*), and UBR5 a second HECT domain-containing E3-ligase with functions in both DNA repair and transcription (*Cojocaru et al., 2011*; *Gudjonsson et al., 2012*). NEURL4, a known HERC2-interacting protein (*Al-Hakim et al., 2012*), was also found along with the de-ubiquitinase enzymes USP7 and USP11 that were preferentially represented in the SNE. In contrast, USP13 shown to regulate MITF stability (*Zhao et al., 2011*) was not detected in our experiments. Several interactions were verified by immunoblot as HERC2, BRG1, USP7, USP11, XRCC5, and XRCC6 were detected in the F-H-MITF immunoprecipitations from the soluble nuclear fraction, but not in the untagged controls (*Figure 1C*), with enrichment of USP7 and USP11 in the SNE compared to the CAE and the selective presence of HERC2 in the SNE.

We identified several other complexes interacting with MITF. Four subunits of TFIIIC, a RNA polymerase III cofactor were detected along with the SMCA1, SMC3, STAG2, and PDS5 cohesin subunits. TRIM28 (TIF1β, KAP1), a co-repressor belonging to a sub-family of TRIM proteins comprising TRIM24 (TIF1α) and TRIM33 (TIF1γ) was found as were the HDAC1 and HDAC2 and the HP1 proteins known to associate with TRIM-co-repressor complexes (*Herquel et al., 2011*). We identified TRRAP, RUVBL1, RUVBL2, and BAF53A. These proteins are already known to form a cofactor for MYC (*Park et al., 2001*, *2002*; *Murr et al., 2007*). Interestingly, a mutation in TRRAP has recently been associated with human melanoma (*Wei et al., 2011*) that together with its interaction with MITF suggests a role in melanomagenesis.

In addition to transcription complexes, the RFC1, RFC2, RFC4, and RFC5 subunits of DNA replication factor C associate with MITF along with the MCM3, MCM5, and MCM7 subunits of the MCM complex that forms at DNA-replication origins. AKAP8 and AKAP8L were also identified and AKAP8 has been shown to interact with the MCM complex (*Eide et al., 2003*), although these protein kinase A anchoring proteins have additional functions (*Collas et al., 1999*). The kinase PLK1 was found in the chromatin-associated fraction suggesting that it may phosphorylate MITF on chromatin. Finally, we detected multiple subunits of the nuclear pore complex. This may reflect control of MITF sub-cellular localization or a coupling of MITF-driven transcription and RNA-export. Together, these data describe a comprehensive set of MITF-interacting proteins.

## MITF interacts with a BRG1-CHD7-containing PBAF complex

While BRG1 was reported to interact with MITF (*de la Serna et al., 2006*; *Keenen et al., 2010*), the composition of the complex had not been determined. We identified BRG1 (SMARCA4) as well as the PBRM1 (BAF180), SMARCC2 (BAF170), SMARCD2 (BAF60B), and ACTL6A (BAF53A) subunits along with CHD7 reported to associate with BRG1 in human neural crest cells (*Bajpai et al., 2010*) (*Figure 1D*). The MITF interacting complex most closely resembles the PBAF variant with the presence of PBRM1 (*Trotter and Archer, 2008*; *Reisman et al., 2009*). To investigate BRG1 complex composition in 501Mel cells, extracts were precipitated with an anti-BRG1 antibody showing co-precipitation of BRG1 with BAF200, BAF155, BAF53A, BAF180, and BAF170, whereas BAF60B and BAF250B were not co-precipitated (*Figure 2A*). Peptides for BAF60B were detected in the MITF interactome, but this subunit did not associate with BRG1 in 501Mel cells, while BAF60A co-precipitated with BRG1, suggesting that MITF interacts with BAF60B independently from the BRG1 complex. The MITF-interacting complex also comprises CHD7 as seen by the reciprocal co-precipitation of BRG1 and CHD7 and their co-precipitation with MITF (*Figure 2A–B*). Together, these observations show that MITF interacts with a novel form of PBAF complex comprising BRG1 and CHD7 (*Figure 2C*).

## BRG1 regulates an extensive gene expression programme essential for proliferation of melanoma cells in vitro

To address the function of BRG1 in melanoma cells, we performed si/shRNA knockdown. si/shRNA knockdown of BRG1 strongly reduced MITF protein and mRNA levels (*Vachtenheim et al., 2010* and *Figure 3A–B*). The effects of BRG1 silencing on gene expression were therefore very similar to the loss of MITF itself with reduced expression of MITF target genes involved in cell cycle, pigmentation and signalling and activation of several genes of the senescence-associated secretory phenotype (SASP) whose expression is also induced upon MITF-knockdown (*Ohanna et al., 2011*) (*Figure 3B*). BRG1 silencing arrested 501Mel proliferation and cells showed a flattened enlarged morphology with multiple cytoplasmic projections, similar to senescent cells upon MITF knockdown (*Figure 3C* [*Strub*

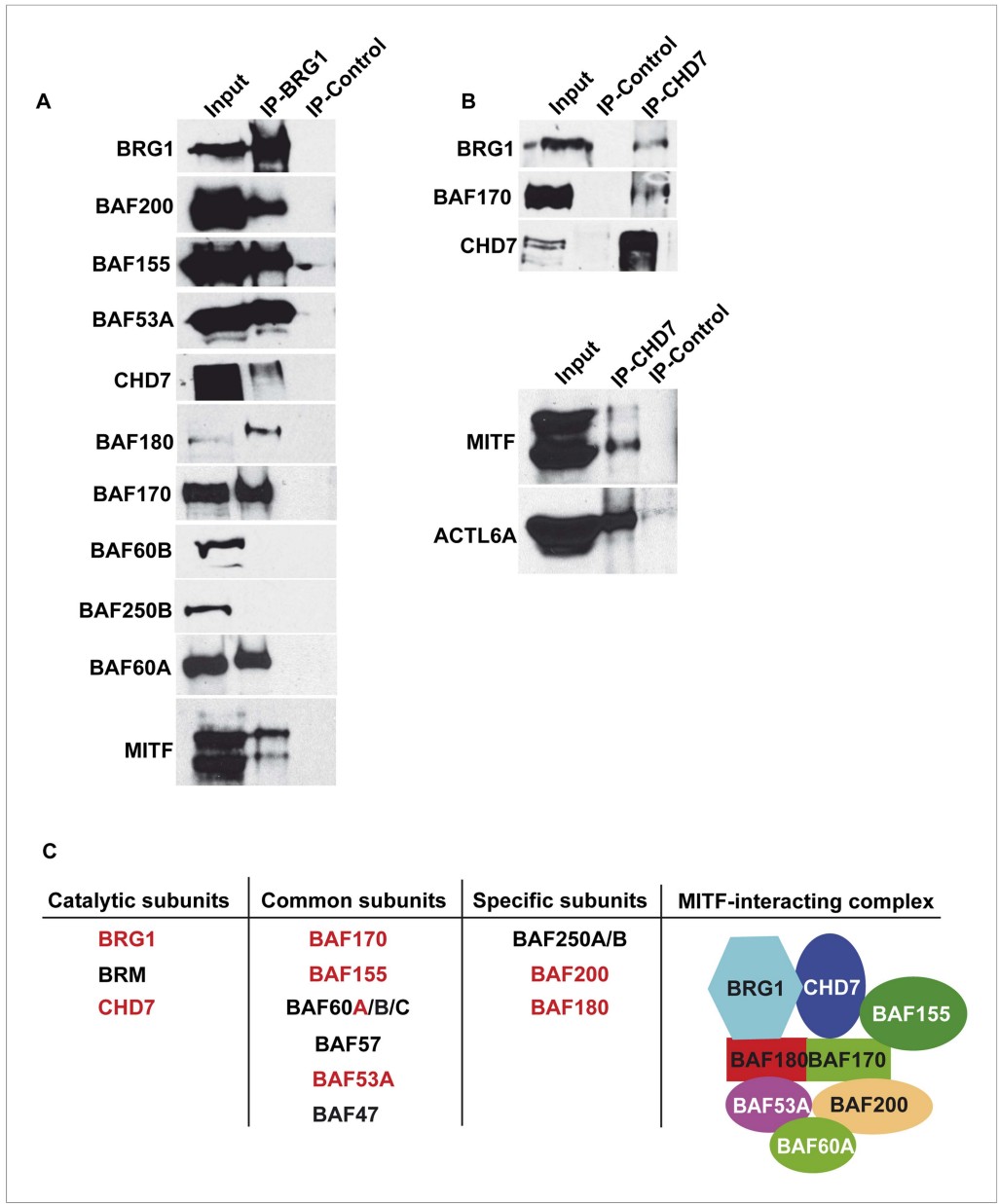

**Figure 2**. Composition of BRG1 complexes in 501Mel cells. (**A**) BRG1 associates with CHD7 in 501Mel cells. Following immunoprecipitation of 501Mel cell extracts with anti-BRG1 antibody or HA beads as control the eluted fractions were probed with antibodies for the indicated proteins. (**B**) Following immunoprecipitation of 501Mel cell extracts with anti-CHD7 antibody or HA beads as control the eluted fractions were probed with antibodies for the indicated proteins. (**C**) Table summarising the known subunits of BRG1-containing complexes highlighting catalytic subunits with ATPase activity, common subunits and specific subunits. The composition of the complex interacting with MITF based on mass-spectrometry and immunoblots is schematised.

*et al., 2011*]). Up to 40% of the BRG1 knockdown cells showed senescence-associated β-galactosidase staining (*Figure 3C*).

RNA-seq following shBRG1 silencing revealed a dramatic effect on gene expression with >4000 genes down-regulated and >5400 genes up-regulated (p.adj <0.05 and log2fold change >1, *Figure 3—figure supplement 1A* and *Supplementary file 2*). ShMITF knockdown resulted in cell cycle arrest and morphological changes associated with senescence (*Figure 3C*) accompanied by the down-regulation of around 600 genes and up-regulation of 747 genes. 60% of genes down-regulated

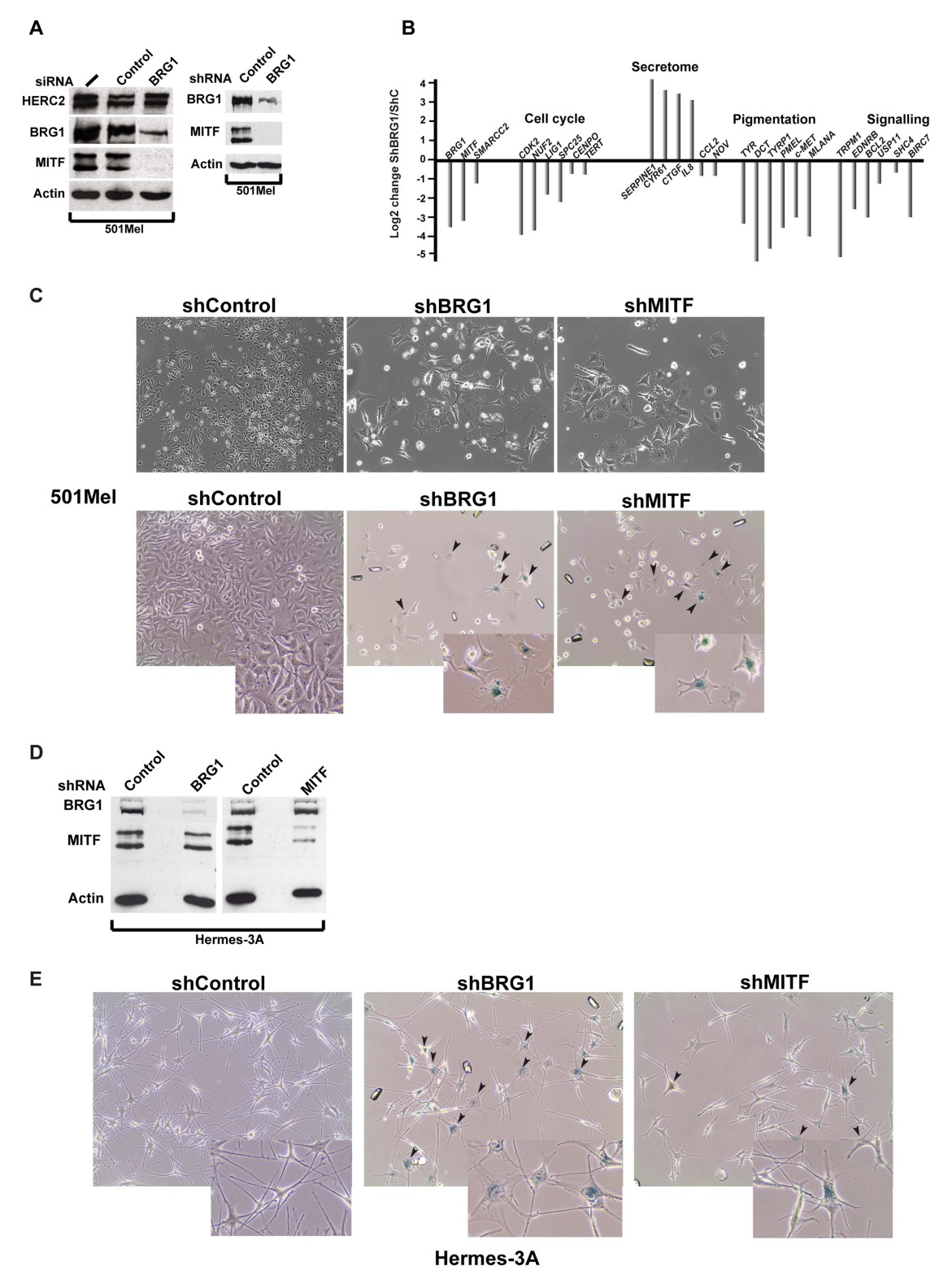

**Figure 3**. BRG1 is essential in melanoma cells and melanocytes. (**A**) Immunoblots of 501Mel cells transfected with control or anti-BRG1 siRNA or cells infected with lentiviral vectors expressing control or anti-BRG1 shRNA. (**B**) Reverse transcription real time qPCR (RT-qPCR) performed on 501Mel cells infected with lentivirus vectors expressing control (**C**) or anti-BRG1 shRNA. The ratio of expression of the indicated genes is shown. (**C**) Phase contrast microscopy of 501Mel cells infected with the indicated shRNA vectors after 5 days of puromycin selection. Magnification X10. The lower panel shows cells
*Figure 3. continued on next page*

*Figure 3. Continued*

stained for senescence-associated β-galactosidase. Arrowheads indicate representative stained cells. Inserts show enlargements of representative cells. (**D**) Immunoblots of Hermes 3A extracts following infection with lentiviral vectors expressing control, anti-BRG1, or anti-MITF shRNAs. (**E**) Phase contrast microscopy of Hermes 3A cells infected with the indicated shRNA vectors after 5 days of puromycin selection and stained for senescence-associated β-galactosidase. Arrowheads indicate representative stained cells. Inserts show enlargements of representative cells.

The following figure supplements are available for figure 3:

**Figure supplement 1**. Gene expression changes in 501Mel and Hermes 3A cells.

**Figure supplement 2**. SOX10 regulates MITF expression in 501Mel cells.

by shMITF showed similar loss of expression upon shBRG1 consistent with the fact that MITF was strongly repressed in the shBRG1 cells (*Figure 3—figure supplement 1A* and *Supplementary file 2*).

Commonly down-regulated genes were enriched in ontology terms associated with signalling, cell cycle, and mitosis (*Figure 3—figure supplement 1A*). Common up-regulated genes were enriched in terms associated with angiogenesis, adhesion, and migration consistent with the dramatic changes in cell morphology. MITF silencing has been shown to induce a SASP comprising *CCL2*, *CTGF* and *SERPINE1* (*Strub et al., 2011*). RNA-seq identified a putative SASP in shMITF cells comprising around 20 secreted factors and of these 15 were also induced in the shBRG1 cells, although several key factors such as *IL8* and *CCL2* were not induced upon BRG1 silencing (*Figure 3—figure supplement 1A*). Loss of either BRG1 or MITF therefore induced senescence of 501Mel cells.

SOX10, TCF/LEF/CTNNB1 and CREB have been reported to activate MITF expression (*Goding, 2000*). We noted that SOX10 expression is strongly repressed in BRG1 knockdown cells, but not in MITF-knockdown cells (*Supplementary file 2*). SiSOX10 silencing repressed endogenous MITF expression (*Figure 3—figure supplement 2A–B*). In 501Mel-Cl8 cells constitutively expressing 3HA-tagged MITF from the CMV promoter (*Strub et al., 2011*), siSOX10 repressed endogenous, but not ectopic MITF. In contrast, siCREB silencing had no effect on MITF expression. SOX10 is therefore a major regulator of MITF expression in 501Mel cells and its diminished expression upon BRG1 knockdown partly explains the concomitant MITF loss. These observations are also consistent with previous reports showing that SOX10 promotes melanoma cell proliferation and that its loss leads to senescence (*Cronin et al., 2013*).

To determine whether the shared phenotypes of BRG1 and MITF knockdown cells resulted from the concomitant loss of MITF upon BRG1 silencing or whether BRG1 acts also as an MITF co-factor, we performed shBRG1 silencing in the 501Mel-Cl8 cells. BRG1 knockdown in these cells repressed endogenous MITF expression, but not ectopic 3HA-MITF (*Figure 3—figure supplement 2C*). Nevertheless, BRG1 silencing elicited a phenotype similar to 501Mel cells characterised by arrested proliferation, and morphological changes. Many MITF target genes were similarly repressed by BRG1 silencing in both 501Mel and Cl8 cells, while SASP components were induced (*Figure 3—figure supplement 2D*). Together, these data show that BRG1 is essential for MITF expression and that it acts as a cofactor for MITF since ectopic MITF in the Cl8 cells does not activate target genes expression in its absence.

## BRG1 and MITF regulate gene expression in human melanocytes

We also investigated BRG1 function in untransformed Hermes 3A melanocytes. In contrast to 501Mel cells, shBRG1 silencing had little effect on MITF expression in Hermes 3A cells (*Figure 3D*), but induced changes in cell morphology with up to 80% of cells showing staining for senescence-associated β-galactosidase (*Figure 3E*). Within 8 days, the BRG1 silenced cells detached from the plate. ShMITF silencing in Hermes 3A cells also led to growth arrest and a marked changes in morphology, with flattening, enlargement of the cell body and reduced neurite projections (*Figure 3D*). Despite these changes indicative of senescence, <50% of shMITF-silenced cells showed staining for senescence-associated β-galactosidase. As with shBRG1, MITF silencing led to cells detaching from the plate within 7 days. Thus, both BRG1 and MITF are essential for melanocyte growth, and in their absence cells undergo growth arrest, senescence, and death.

RNA-seq showed that BRG1 silencing in Hermes 3A cells down-regulated 587 genes, and up-regulated 971 genes, many fewer than in 501Mel cells (*Figure 3—figure supplement 1B–C*, and *Supplementary file 2*). Down-regulated genes were involved in pigmentation, cholesterol metabolism as well as intracellular signalling cascades and general cell morphology. Up-regulated genes were involved in cell–cell signalling and adhesion as well as angiogenesis. MITF silencing down-regulated 757 genes and up-regulated 664 genes. Comparisons showed that 38% of genes down-regulated upon BRG1 silencing were diminished by MITF silencing, while 25% of the BRG1 up-regulated genes were also increased upon MITF silencing.

Together these results show that BRG1 is critical for Hermes 3A proliferation despite the fact that it regulated a much reduced gene expression programme in these cells. As with the 501Mel cells however, MITF and BRG1 cooperate to regulate a subset of genes involved in several essential cellular processes associated with resistance to apoptosis, cell morphology, and signalling.

## BRG1 is essential for generation of melanocytes in vivo

As BRG1 is essential for proliferation of both melanocytes and melanoma cells in vitro, we asked if BRG1 is also essential for melanocytes in vivo in mice. To bypass the embryonic lethal phenotype of BRG1 germ-line knockout (*Bultman et al., 2000*), we crossed mice with floxed alleles of the *Smarca4* gene encoding BRG1 (*Indra et al., 2005*) with *Tyr*-Cre mice to allow selective inactivation of BRG1 in the melanocyte lineage (*Delmas et al., 2003*). We first generated *Tyr*-Cre::*Smarca4*$^{lox/+}$ mice that were crossed to generate the resulting *Smarca4*$^{mel+/−}$ or *Smarca4*$^{mel−/−}$ genotypes. The *Tyr*-Cre::*Smarca4*$^{mel−/−}$ mice were present in the progeny at the expected ratio indicating no loss of viability (*Figure 4A*). While the heterozygous *Tyr*-Cre::*Smarca4*$^{mel+/−}$ mice were black, homozygous *Tyr*-Cre::*Smarca4*$^{mel−/−}$ mice were either completely white or had occasional black spotting presumably arising from rare melanoblasts that had escaped Cre-driven recombination during embryogenesis (*Figure 4B*). Examination of the hair from two genotypes showed an absence of melanin in *Smarca4*$^{mel−/−}$ hair.

We labelled the hair follicles of the *Smarca4*$^{mel+/−}$ or *Smarca4*$^{mel−/−}$ animals with antibodies against Dct and Sox10. In *Smarca4*$^{mel+/−}$ mice, Sox10 stained the nucleus of mature bulb melanocytes whose cytoplasm was stained with Dct (*Figure 4C*). In contrast, there was no staining with these antibodies in *Smarca4*$^{mel−/−}$ bulbs. There were therefore no identifiable mature melanocytes in the mutant animals. Taken together with the essential role of BRG1 in melanoma and melanocyte cells in vitro, these data indicate that BRG1 is essential for generation of mature melanocytes in vivo.

## BRG1 binds widely over the 501Mel cell genome

Given the critical function of BRG1 in the melanocyte lineage in vitro and in vivo, we used ChIP-seq to better understand how MITF and BRG1 regulate gene expression in 501Mel cells. We performed BRG1 ChIP-seq on native Mnase-digested chromatin along with a GFP-control ChIP-seq and identified >42,400 BRG1-occupied sites located in inter and intragenic regions, but with relative enrichment at the promoter (*Figure 5A*). Comparison with public ENCODE data showed BRG1-occupied sites often co-localized with or flanking enhancer regions marked with acetylated H3K27(ac) a mark of active enhancers (*Smith and Shilatifard, 2014*) (*Figure 5—figure supplement 1A–B*). As melanocyte-specific enhancers are not represented in public ENCODE data, we integrated the BRG1 profile with public H3K27ac data from primary human foreskin melanocytes and with H3K27ac data from a proliferative primary melanoma culture (*Verfaillie et al., 2015*). BRG1 co-localized with enhancers active in melanocytes (*Figure 5—figure supplement 1A–B*) including melanocyte-specific enhancers seen at the MITF and DCT loci (see below). The primary melanocyte and proliferative melanoma H3K27ac profiles are very similar and BRG1 co-localized with around 15,000 melanocyte/melanoma H3K27ac marked regions at promoters as well as distal inter and intragenic enhancers (*Figure 5B*). Analysis of DNA sequence motifs at BRG1 peaks identified binding sites for a large set of transcription factors (TFs), the most abundant of which are NRF1, KLF5, SP1-family factors, CTCF, EGR1, and E-boxes (*Figure 5—figure supplement 1C*). Thus, BRG1 is recruited by many different TFs to its sites on the 501Mel genome.

BRG1 occupied sites displayed a variety of profiles. A set of highly occupied sites (Clusters A–C, *Figure 5C*) comprised two peaks separated by various distances. Re-clustering of each of these sets revealed a complex profile comprising in general two more or less sharply defined peaks separated by distances of 450–800 bp (*Figure 5D*). At other sites however, BRG1 occupied a single sharp peak (cluster D, *Figure 5C*).

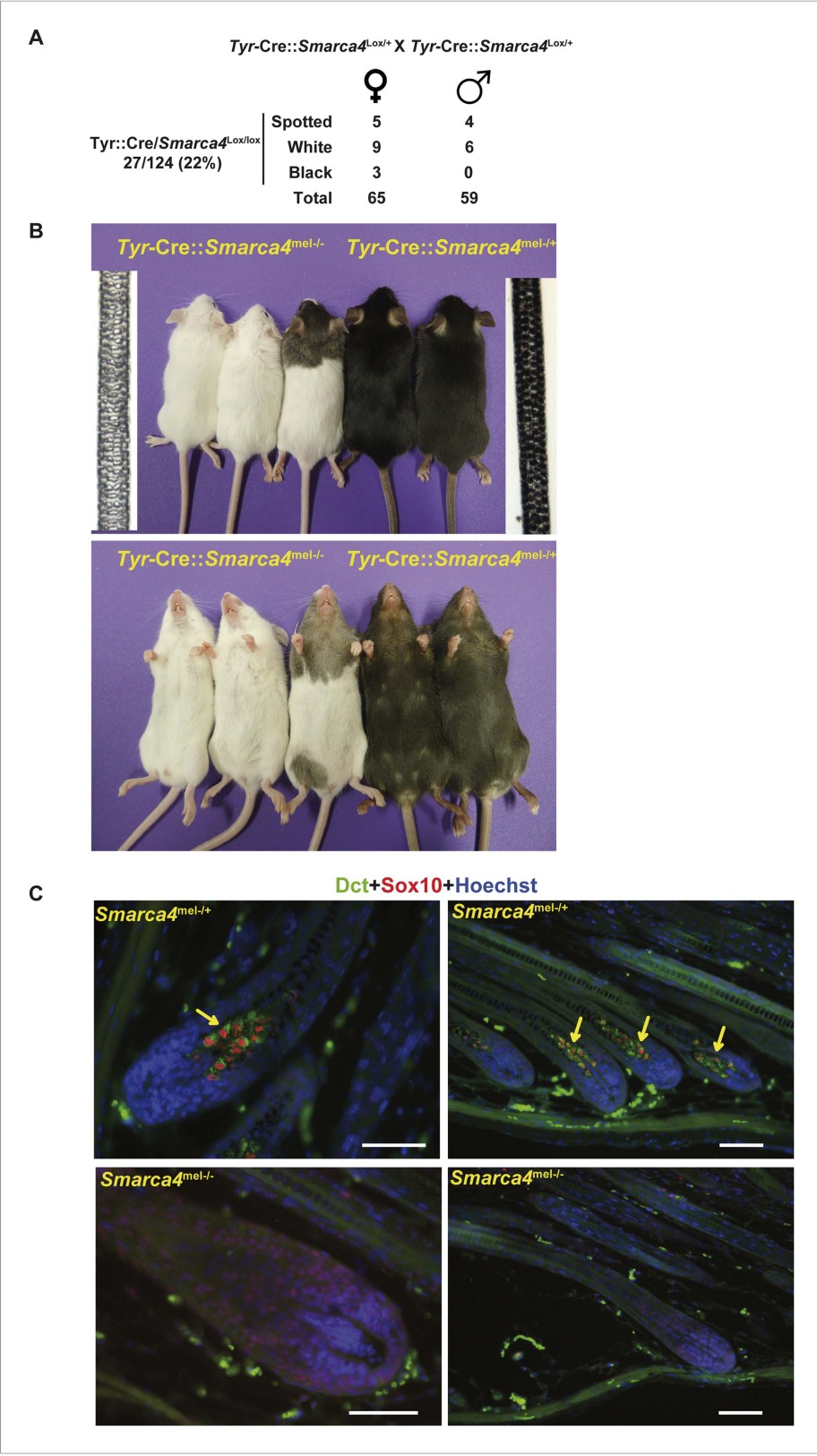

**Figure 4.** BRG1 is essential in mouse melanocytes in vivo. (**A**) Statistics relevant to the phenotype of the mice lacking Brg1 in the melanocyte lineage. Note that as the *Tyr*-Cre transgene is present on the X chromosome, we detect black mice with the *Tyr*-Cre::*Smarca4*^lox/lox genotype, but in these animals the Cre-recombinase is subjected to X-
*Figure 4. continued on next page*

*Figure 4. Continued*

inactivation such that recombination does not occur in all melanoblasts. (**B**) Photographs of representative mice of the indicated genotypes. Bright field images of hair from the backs of 8-week-old animals of the two genotypes are also shown. Magnification X40. (**C**) Labelling of dorsal hair follicle bulbs with antibodies against Sox10 in red and Dct in green along with Hoechst-stained nuclei in blue. Arrows indicate labelled melanocytes in the bulb of Brg1 expressing mice. Scale bars are 50 μm.

BRG1-occupied sites at the TSS also displayed different profiles. BRG1 was preferentially located either upstream or downstream of the TSS (*Figure 5—figure supplement 1D*, clusters A–D) or showed equivalent occupancy at both sites (cluster E), or was absent (clusters F–G). Re-analysis of cluster E revealed BRG1 localized at varying distances between −370 and +440 bp with respect to the TSS (*Figure 5E*, clusters A–F), although it was present only on the downstream nucleosome in cluster G. BRG1 therefore appears to bind to the last (−1) nucleosome before the nucleosome-depleted region (NDR) at the TSS and the first downstream (+1) nucleosome although the distance between these nucleosomes is variable as previously described (*Fenouil et al., 2012*). The metaprofile of all BRG1-occupied TSS showed preferential location of BRG1 at −200 bp and +72 bp relative to the TSS (*Figure 5F*). Integration of RNA Polymerase II (Pol II) ChIP-seq data from 501Mel cells showed the peak of promoter paused Pol II located immediately upstream of the BRG1-bound +1 nucleosome. BRG1 therefore binds a large subset of sites both at the TSS and non-TSS regions as paired BRG1-bound nucleosomes separated by variable distances.

Annotation of BRG1-occupied sites identified 7168 potential target genes in a window of ±10 kb around the TSS and >10,000 potential target genes in a window of ±30 kb (*Figure 5—figure supplement 1E*). Integration with RNA-seq data showed that 34% of genes down-regulated upon BRG1 silencing had at least one BRG1-occupied site at ±10 kb, while this figure attained 47% using the ±30 kb window. In contrast, only 12% and 18% of up-regulated genes were associated with BRG1-occupied sites using these windows. BRG1-associated and down-regulated genes were enriched in pigmentation, DNA replication, and glucose and fatty acid metabolism (*Figure 5—figure supplement 1F*). BRG1-associated up-regulated genes were involved in angiogenesis and transcription regulation, notably expression of a set of ZNF TFs that appear to be co-ordinately regulated by BRG1 (*Supplementary file 3*). Together these data show that BRG1 is recruited by many TFs in 501Mel cells to regulate the expression of a large set of target genes.

## MITF and SOX10 co-localization with BRG1

If BRG1 acts as a cofactor for MITF, they should co-localize on the genome. Integration of BRG1 and MITF ChIP-seq data showed that of >16,000 MITF occupied sites, 5867 were co-occupied by BRG1 and were located in both inter and intragenic regions, but enriched at promoters (*Figure 6A*). Co-localization can be observed at the *MITF* locus itself, where MITF occupied multiple sites, several of which co-localized with BRG1 and melanocyte H3K27ac-marked enhancers throughout this locus (*Figure 6B*). A similar situation was seen at the DCT locus, where MITF and BRG1 co-localized at H3K27ac marked melanocyte-specific enhancers.

At co-occupied sites, MITF binds between two BRG1 peaks as observed at the *TYR* locus (*Figure 6C*). BRG1 peaks were either positioned almost symmetrically with respect to MITF or displayed shoulders of density either upstream or downstream presumably corresponding to additional, but less well-positioned BRG1-occupied nucleosomes (*Figure 6C*). Genes associated with co-occupied sites are involved in previously defined critical functions of MITF such as lysosome biogenesis (*Ploper et al., 2015*), pigmentation, cell cycle, apoptosis, and DNA damage response (*Figure 6—figure supplement 1A*, and *Supplementary file 4*). Co-occupied sites showed enrichment not only in E and M-boxes for MITF, but also in motifs for SOX10, CREB, and ETS1 (*Figure 6—figure supplement 1B*), TFs that may cooperate with MITF and BRG1 to regulate associated target genes. In addition, we also identified a set of 1065 TSS, where BRG1 occupies the nucleosomes flanking the TSS and MITF binds the NDR (*Figure 6D*). Genes associated with these promoters were involved in lysosome formation, cell cycle, and RNA processing and splicing (data not shown).

As the SOX10 motif was so highly represented at MITF-BRG1 sites and that it physically interacts with BRG1 complexes (*Weider et al., 2012*), we performed SOX10 ChIP-seq in 501Mel cells to profile

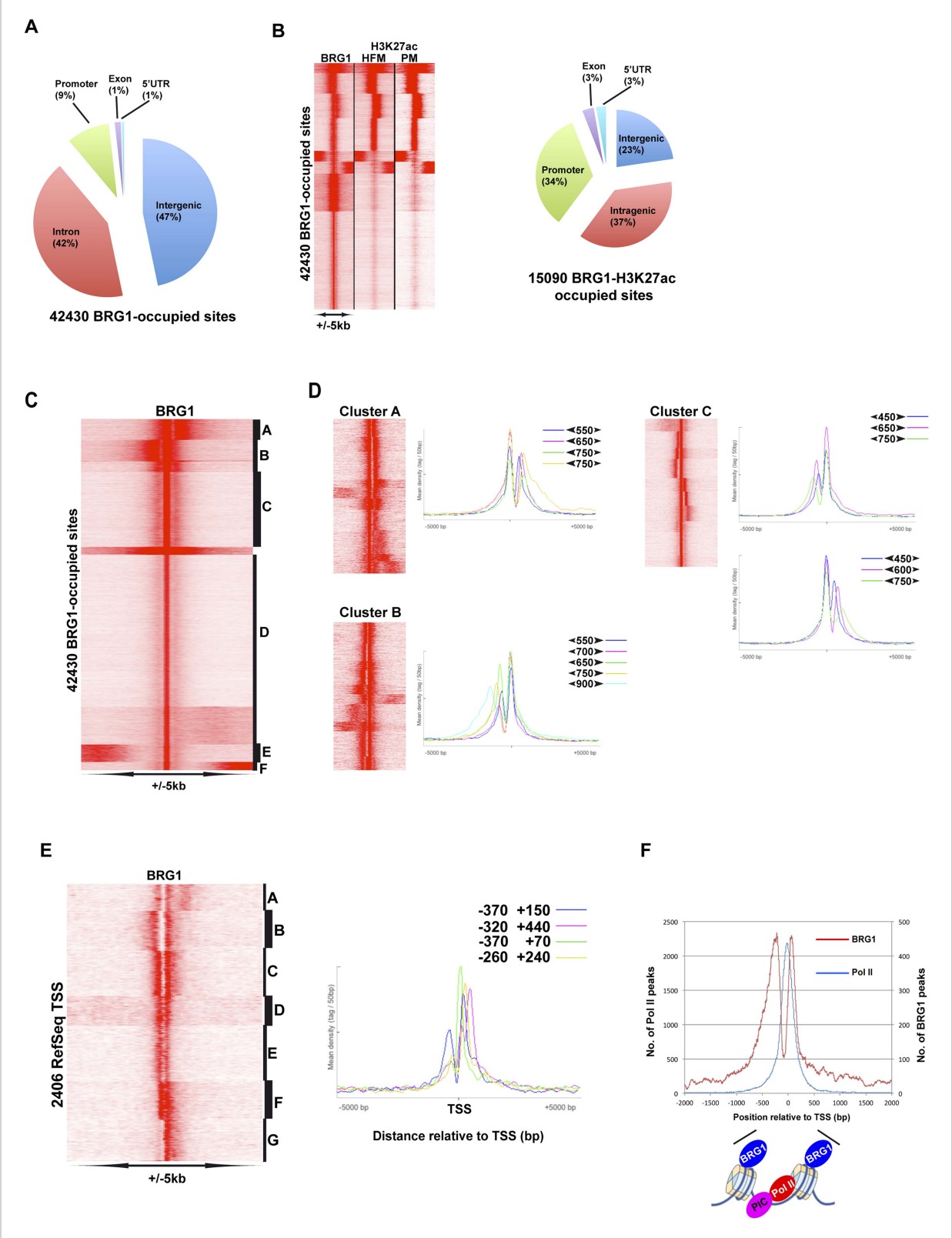

**Figure 5**. Genome-wide BRG1 occupancy. (**A**) Pie chart showing distribution of BRG1-bound sites with respect to genomic features. (**B**) Clustering of BRG1 occupied sites with primary melanocyte and proliferative melanoma H3K27ac-marked elements and their distribution with respect to genomic features. (**C**) Read density clustering of BRG1 at its occupied sites reveals several profiles of BRG1 occupancy. (**D**) Clusters A–C from panel **C** were

*Figure 5. continued on next page*

*Figure 5. Continued*

re-clustered and representative meta-profiles are shown to illustrate the distances separating the BRG1 peaks with the large and small arrowheads indicating the stronger and weaker peaks, respectively. (**E**) Re-analysis of RefSeq TSS where BRG1 is present reveals different localizations of BRG1-bound nucleosomes. Several representative meta-profiles of different subclasses are shown. The peak coordinates in each class relative to the TSS are indicated. (**F**) The overall meta-profile of BRG1 TSS occupancy is superimposed on the RNA Pol II meta-profile. The binding profiles of BRG1 and Pol II around the TSS are schematised below the graph.

The following figure supplement is available for figure 5:

**Figure supplement 1**. Profiling of BRG1 genome occupancy.

its genomic occupancy. Almost 6000 SOX10-occupied sites were detected, but unlike MITF they were mainly located at distal regulatory elements with only around 80 sites at the proximal promoter. (*Figure 6—figure supplement 2A*). SOX10 recognition motifs were strongly enriched at SOX10 bound sites some of which were organised as degenerate palindromes (*Figure 6—figure supplement 2B*). Recognition motifs for TCF/LEF, MITF, and TFAP2A were also enriched at these sites. Genes associated with these sites were enriched in several ontology terms associated with cell motion/adhesion, cell morphogenesis and organisation (*Figure 6—figure supplement 2C*). We integrated the SOX10 data with that of MITF, BRG1, and H3K27ac. SOX10 and MITF co-occupied 3674 sites where MITF was located either 3′ or 5′ to SOX10 (*Figure 6—figure supplement 2D*). SOX10, MITF, BRG1, and H3K27ac were found at 1929 sites (*Figure 7A*, clusters A–D) while varying levels of SOX10, MITF and BRG1 were found at an additional ≈2000 sites (cluster E), whereas SOX10 and BRG1 co-occupied 1159 sites (cluster F) and 972 sites displayed essentially only SOX10 (cluster G). The above data define a specific organisation of MITF-associated regulatory elements (MAREs) where MITF alone or with SOX10 bind between two BRG1-bound nucleosomes. This analysis also defined 1929 MAREs at active melanocyte/melanoma enhancer elements associated with 1511 genes involved in pigmentation, cell motility and adhesion, actin cytoskeleton organisation, and signalling (*Supplementary file 4*).

## Co-localization of MITF and BRG1 with YY1 and TFAP2A

We also found a large overlap between BRG1 and YY1 genome occupancy (*Figure 7—figure supplement 1A*). Despite the fact that YY1 ChIP-seq was performed in a distinct melanoma line (*Li et al., 2012*), we identified 2853 sites where YY1 co-localized with MITF and 3060 SOX10-YY1 co-occupied sites (*Figure 7B*). All three proteins co-localize at 1070 sites together with BRG1 and 530 of these sites are marked with H3K27ac (*Figure 7C*). These data define a set of MAREs comprising YY1, BRG1, MITF, and SOX10 located at active melanocyte/melanoma enhancers. These 530 MARES are associated with genes involved in pigmentation, melanocyte development, cell motion as well as apoptosis (data not shown).

TFAP2A is a TF involved in melanoma (*Huang et al., 1998*) and normal melanocyte function (*Brewer et al., 2004*; *Van Otterloo et al., 2010*) and co-localizes with MITF at promoters involved in pigment cell differentiation in primary human melanocytes (H Seberg, E van Otterloo, and RA Cornell, manuscript in preparation). In addition, TFAP2A binding motifs are enriched at BRG1 bound sites (*Figure 5—figure supplement 1B*). In agreement with this, of the 13,693 TFAP2A-bound sites in human melanocytes, 6432 sites co-localized with BRG1 and H3K27ac defining a large set of active TFAP2A-bound regulatory elements (*Figure 7—figure supplement 1B*). Also, at more than 1000 sites, MITF was located at varying distances 5′ or 3′ to TFAP2A, and YY1 and TFAP2A colocalized at 4819 sites (*Figure 7—figure supplement 1C* and data not shown). Integrative analysis identified 762 YY1-TFAP2A sites also occupied by BRG1 and MITF and marked by H3K27ac, defining a set of TFAP2A-YY1-containing MAREs (*Figure 7—figure supplement 1D*). Analysis of the genes associated with these MAREs showed enrichment in those involved in lumen organisation, melanosomes, and transcriptional regulation (data not shown).

## Genome-wide recruitment of BRG1 by MITF and SOX10

To determine whether MITF and/or SOX10 actively recruit BRG1 to the genome, we performed BRG1-ChIP-seq following siRNA-mediated silencing of MITF, SOX10 or siLuciferase (Luc) as control. At the *TYR* locus, siMITF silencing led to the specific loss of BRG1 at a promoter proximal site (* in

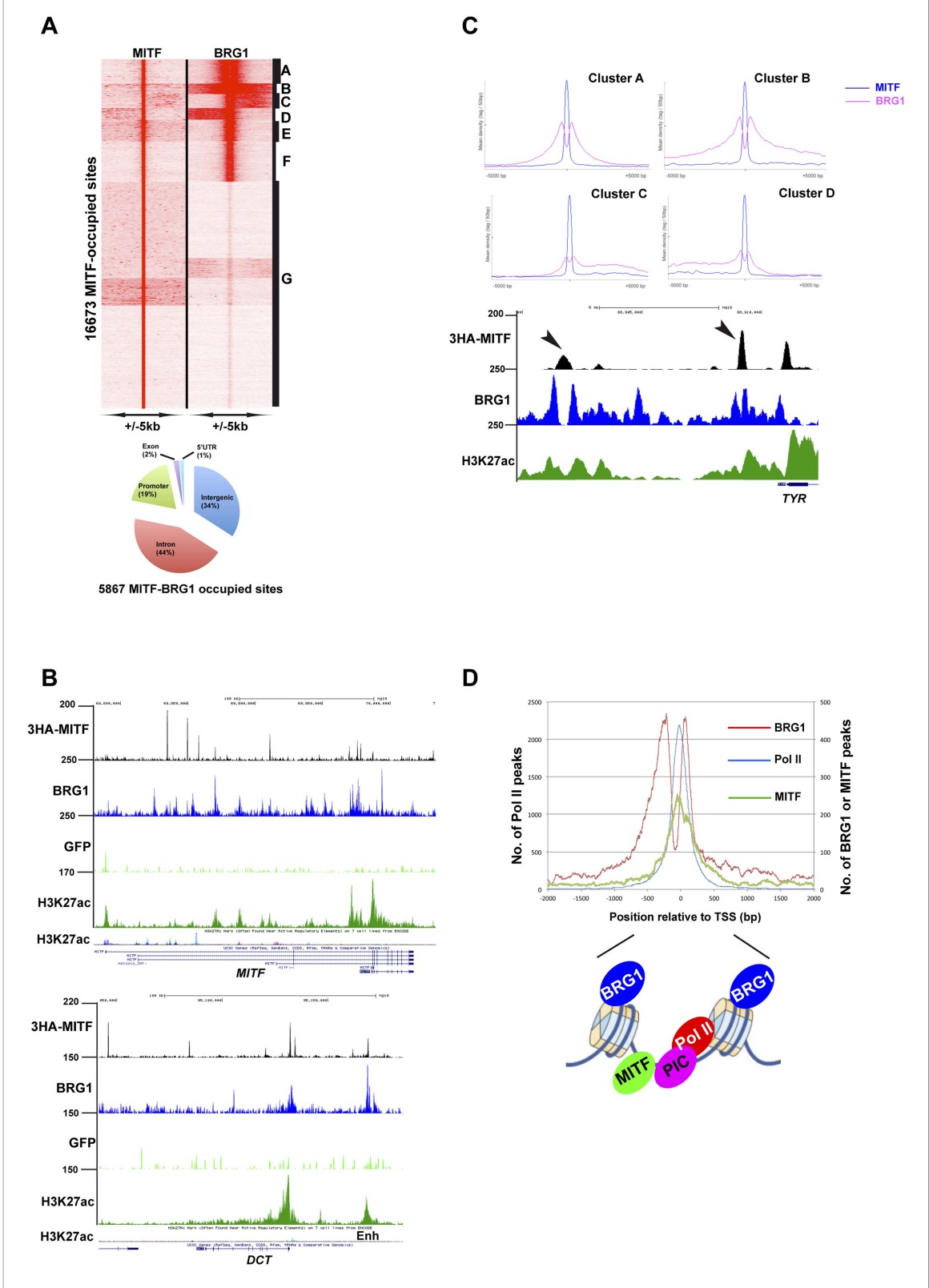

**Figure 6**. Sites co-occupied by BRG1 and MITF. (**A**) Read density clustering of BRG1 and MITF at MITF-occupied loci reveals co-occupied sites with different profiles and their distribution with respect to genomic features. (**B**) UCSC screenshots showing MITF and BRG1 occupancy over the *MITF* and *DCT* loci highlighting their co-localization along with that of melanocyte-specific H3K27ac-marked enhancers. A GFP ChIP was performed as a negative

*Figure 6. continued on next page*

*Figure 6. Continued*

control. (**C**) Meta-profiles of BRG1 occupancy in clusters A–D of panel **A**. The lower panel shows a UCSC screenshot of MITF and BRG1 occupancy over the *TYR* locus highlighting the binding of MITF between two BRG1-occupied nucleosomes. (**D**) Meta-profile showing occupancy by BRG1, MITF, and Pol II at the TSS. The binding profiles of each factor BRG1 and Pol II around the TSS are schematised below the graph.

The following figure supplements are available for figure 6:

**Figure supplement 1**. TF binding motifs at MITF and BRG1-bound sites.

**Figure supplement 2**. MITF co-localizes with SOX10.

*Figure 8A*), whereas at the upstream site where MITF co-localized with SOX10 little change was observed (arrowheads in *Figure 8A*). In contrast, siSOX10 and consequent loss of both SOX10 and MITF (see above), diminished BRG1 at all sites. Interestingly, BRG1 occupancy was observed throughout the region between these sites. siSOX10 silencing led to BRG1 loss across the entire region suggesting that it is recruited to the SOX10/MITF sites and then may spread across the locus. A similar situation was observed at the *CIT* locus encoding a MITF-regulated gene essential for cell cycle progression, with two downstream regulatory elements comprising MITF and SOX10 sites (* and arrowheads in *Figure 8B*). At both sites, BRG1 was strongly reduced upon siMITF or siSOX10 silencing. A global analysis identified sites showing a strong and preferential loss of BRG1 following, siSOX10 (clusters B and G–H, *Figure 8C*), siMITF (clusters E and F) or both, (cluster D), defining sites where MITF and SOX10 independently or cooperatively recruit BRG1 (*Figure 8D*). These data indicate that MITF and SOX10 actively recruit BRG1 either cooperatively or specifically to a large number of genomic loci.

While many genes associated with MAREs were down-regulated upon MITF silencing, MAREs may also be involved in gene silencing. Expression of SERPINE1 and IL24, two SASP components, was strongly up-regulated in senescent siMITF cells (see above). At both loci, BRG1 and MITF were present at either the promoter and/or distal regulatory regions and were lost upon siMITF coinciding with activation of their expression (*Figure 8—figure supplement 1A–B*). These observations suggest that BRG1 recruitment by MITF at these genes acts to silence their expression. Furthermore, BRG1 was re-localized over the genome to new sites during this process for example to *CCL2* that was strongly induced upon siMITF silencing (*Figure 8—figure supplement 1C*).

We also noted that different sequence motifs were enriched at MITF-bound sites associated with either up or down regulated genes (*Figure 6—figure supplement 1C–D*). In particular, we observed that nuclear receptor half sites were enriched at loci associated with both classes. The finding of nuclear receptor half sites associated with MITF is unexpected as little is known about nuclear receptor function in melanoma although a therapeutic role of LXRb agonists has been demonstrated (*Pencheva et al., 2014*). Nuclear receptors can therefore be added to the list of factors that may interact functionally with MITF.

We next investigated whether on the other hand MITF occupancy is affected by BRG1 silencing using Cl8 cells where BRG1 loss did not affect expression of ectopic 3HA-tagged MITF (see above). Anti-HA ChIP-qPCR at MITF sites not associated with BRG1 co-occupancy showed only a mild reduction in MITF occupancy (*Figure 9A*, group A). In contrast, at many co-occupied sites, BRG1 silencing resulted in a marked increase in MITF occupancy. This was observed for example at the *GPR110*, *UVRAG*, and *SOX6* loci (group B), where MITF binds to sites located between BRG1-bound nucleosomes (*Figure 9B*). Increased occupancy is not however seen at all co-occupied sites since at the *DCT* and *SOX10* loci MITF occupancy was not affected (group C). Thus, BRG1 regulates the dynamics of MITF binding to a subset of sites

## Discussion

### A complex network of protein interactions around MITF

A comprehensive characterisation of the MITF interactome revealed its association with multiple complexes, including PBAF and NURF, TFIIIC, cohesins and multiple enzymes of the ubiquitin cycle.

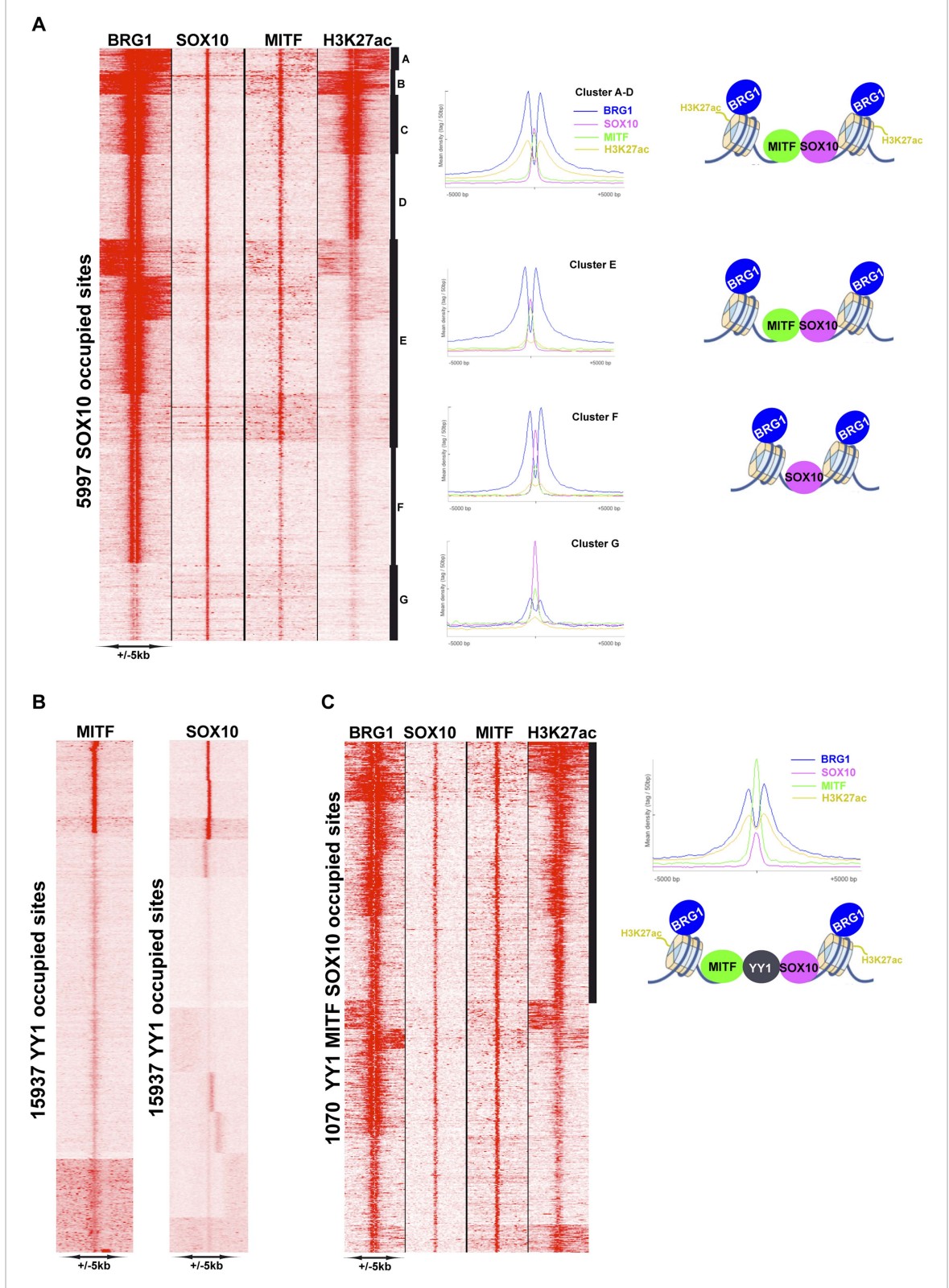

**Figure 7**. MITF, SOX10, and YY1 co-localize with BRG1. (**A**) Read density clustering illustrating regions of co-localization of the indicated factors and melanocyte H3K27ac. The meta-profiles for the indicated clusters are shown to the right illustrating the binding of MITF and SOX10 between the two BRG1-bound and H3K27ac-marked nucleosomes. The binding profiles are schematised on the right of the figure. (**B**) Co-localization of MITF and SOX10

*Figure 7. continued on next page*

*Figure 7. Continued*

on YY1-occupied sites. (**C**) Similar to panel **A**, read density clustering illustrates MAREs with co-localization of H3K27ac, BRG1, MITF, and SOX10 at YY1-bound regions. A meta-profile is shown to the right along with a schematic representation of these sites.
The following figure supplement is available for figure 7:

**Figure supplement 1**. Identification of YY1-TFAP2A associated MAREs.

TFE3, TFEB, and TFEC also co-purify with MITF raising the question of which sites are bound as MITF homodimers or MITF-TFE heterodimers and whether binding as homo- or heterodimers has functional consequences for transcription regulation. Mouse genetic studies have not revealed functions for the TFE factors in development of the melanocyte lineage (*Steingrimsson et al., 2002*), however their role in melanoma has not been fully evaluated.

A striking finding of our study is the interaction of MITF with multiple proteins involved in DNA damage and repair, suggesting that MITF may influence these processes. Nevertheless, we so far found no recruitment of MITF to DNA damage sites (unpublished observations), the significance of these interactions therefore remains to be determined. Similarly, MITF associates with TFIIIC suggesting that it may regulate Pol III transcription. Alternatively, the presence of cohesin subunits in the interactome suggests that MITF may associate with TFIIIC at 'Extra TFIIIC' sites that organise chromatin structure in association with the cohesin complex (*Noma et al., 2006*; *Kirkland et al., 2013*).

MITF also interacts with complexes involved in DNA replication. Assembly of DNA replication origins can be influenced by the presence of transcription regulatory elements and MYC that also interacts with MCM subunits, has been shown to directly influence DNA replication (*Dominguez-Sola et al., 2007*; *Méchali, 2010*). Perhaps MITF interaction with these replication complexes facilitates replication origin assembly at MITF-occupied sites thereby coupling replication with transcription regulation in a melanocyte-specific manner.

## A novel PBAF complex containing BRG1 and CDH7 is a cofactor for MITF

MITF interacts with BRG1, but the related Brahma (BRM) protein was not detected, although both proteins are expressed in 501Mel cells (*Keenen et al., 2010*). MITF interacts with a PBAF-like complex comprising CHD7. We have so far been unable to co-localize BRG1 and CHD7 by ChIP-seq due to the lack of ChIP-seq grade CHD7 antibodies (our unpublished data).

In 501Mel cells, BRG1 regulates an extensive gene expression programme including MITF and SOX10 whose expression is lost upon BRG1 silencing and cells undergo growth arrest and senescence characterised by a SASP similar to that seen upon MITF silencing. This requirement for BRG1 to drive a large gene expression programme required for melanoma cell proliferation in vitro contrasts with its tumour suppressor function in human cancers including melanoma (*Hargreaves and Crabtree, 2011*; *Shain and Pollack, 2013*; *Bastian, 2014*; *Wang et al., 2014*).

BRG1 is also essential in Hermes-3A cells and the overlapping gene expression changes suggest that BRG1 acts as an MITF cofactor in these cells. Nevertheless, while the impact of BRG1 silencing is less extensive than in 501Mel cells the changes in gene expression lead to senescence and eventual cell death. As BRM is expressed in Hermes-3A cells, there is a potential for redundancy between BRG1 and BRM that could account for the less extensive effect, although it is important to note that this is not the case in 501Mel cells. BRG1 knockout in developing mouse melanocytes in vivo leads to complete loss of pigmentation and immunostaining failed to reveal the presence of Sox10 or Dct expressing cells in the hair follicle of the mutant animals. The absence of expression of these important markers indicates that there are no identifiable melanocytes in the *Smarca4*[mel−/−] hair follicles. This together with the arrested proliferation and cell death seen in melanoma cells and melanocytes in vitro, indicate that loss of pigmentation most likely results from an absence of melanocytes in these animals. Together our data identify BRG1 as an essential MITF cofactor in melanoma and melanocyte/melanoblast cells in vitro and in vivo.

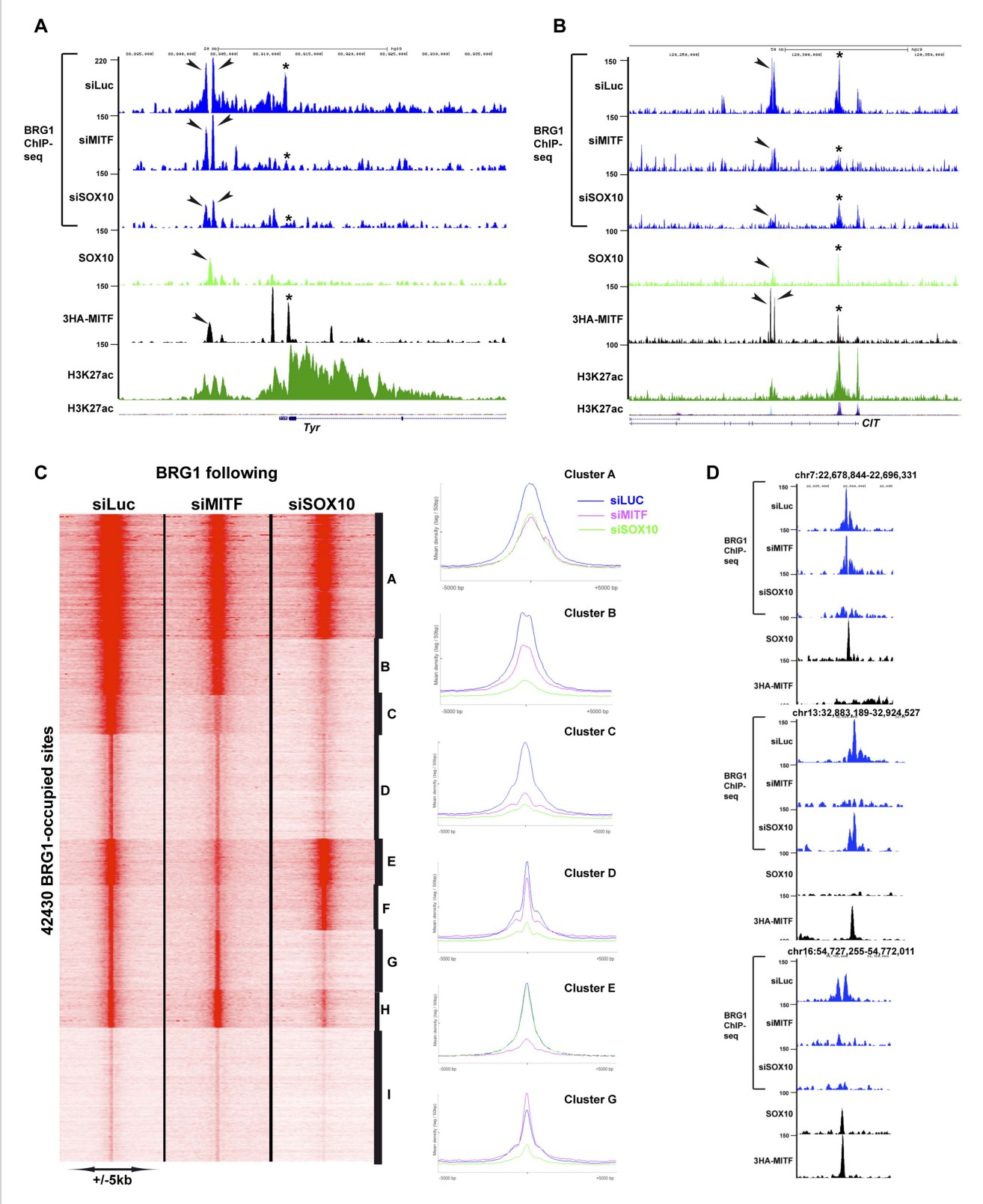

**Figure 8**. MITF and SOX10 actively recruit BRG1. (**A**) UCSC screenshots illustrating occupancy over the *TYR* locus and highlighting the selective recruitment of BRG1 by MITF and/or SOX10. * illustrates a region where BRG1 occupancy is diminished upon siMITF silencing, while the arrow indicates a region where BRG1 is lost upon siSOX10 silencing. (**B**) * illustrates a region where BRG1 occupancy is diminished upon siMITF silencing, while the arrow

*Figure 8. continued on next page*

*Figure 8. Continued*

indicates a region where BRG1 is diminished upon both siMITF and siSOX10 silencing. (**C**) Read density clustering illustrating set of sites whose BRG1 occupancy is diminished following siMITF or siSOX10 silencing compared to siLuc. The meta-profiles of several representative clusters are shown to the right of the figure. (**D**) UCSC screenshots showing examples of BRG1 recruitment by SOX10 or MITF.
The following figure supplement is available for figure 8:

**Figure supplement 1**. BRG1 and MITF repress gene expression.

## BRG1-YY1-SOX10-TFAP2A-MITF co-localization defines sets of MAREs with a specific chromatin organisation

BRG1 binds extensively over the melanoma cell genome often at active H3K27ac-marked enhancers consistent with previous reports (*Hu et al., 2011*). Nevertheless, we describe here a novel profile where BRG1 often binds as two peaks separated by 250–800 base pairs. These BRG1 sites define two classes of elements. The first is a subset of TSS, where BRG1 occupies the nucleosomes flanking the NDR with the Pol II pause site located immediately 5′ of the +1 nucleosome. This specific positioning of BRG1 at the TSS was not noted in several previous ChIP-seq studies (*De et al., 2011*; *Euskirchen et al., 2011*; *Ho et al., 2009*; *Hu et al., 2011*; *Yu et al., 2013*; *Morris et al., 2014*), however, *Tolstorukov et al. (2013)* reported that BRG1 occupies the nucleosomes flanking the TSS. Tolstorukov et al. also reported that inactivation of BRG1 does not affect the positioning of these two nucleosomes, but rather elicits a strong reduction in their occupancy. Nevertheless, the overall meta-profiles presented by Tolstorukov et al., analogous to that reported here, overlooked two important features: (1) that many promoters show BRG1 occupancy of only the −1 or +1 nucleosome, and (2) that −1 or +1 nucleosome location is variable (*Fenouil et al., 2012*).

The second class corresponds to TF binding sites in promoter and enhancer elements. The observed variability in distances separating the BRG1-bound nucleosomes likely reflects the number of TFs bound in the intervening regions. For example, combinations of MITF, SOX10, YY1, TFAP2A as well as other TFs such as ETS1 bind between two BRG1-bound nucleosomes. At many of these sites the BRG1-bound nucleosomes are also marked by H3K27ac, thus defining MAREs active in regulating melanocyte lineage gene expression. As SOX10, YY1, TFAP2A, and ETS1 all have important roles in melanocytes and/or melanoma, the MAREs identified here define a combinatorial signature of TFs critical for gene regulation in this lineage (*Figure 9*). This is further underlined by the recent finding that a combination of MITF, SOX10, and PAX3 can reprogram fibroblasts into functional melanocytes (*Yang et al., 2014*). Ondrusova et al. reported an MITF-independent pro-survival role for BRG1 in melanoma cells (*Ondrušová et al., 2013*). This observation is in agreement with our findings that BRG1 silencing affects expression of many more genes than MITF and that the binding motifs for a variety of factors other than MITF are enriched at BRG1-occupied sites. BRG1 is likely therefore to act as a cofactor for many other TFs accounting for its MITF-independent functions.

Our data are consistent with the idea that MITF and other combinations of TFs bind the DNA between two BRG1-occupied and H3K27ac-marked nucleosomes. Such an organisation has been previously proposed based on extensive ChIP-seq profiling by the Encode consortium showing that TF binding sites are often combinatorial and correspond to GC-rich, DNaseI hypersensitive NDRs flanked by two positioned nucleosomes (*Wang et al., 2012*). Our data support this idea through the identification of combinatorial MAREs and they extend it by showing that the nucleosomes flanking the TF binding sites are often bound by BRG1 and marked with H3K27ac (*Figure 9*). These regulatory elements show an analogous organisation to that of the TSS that also comprise a NDR encompassing variable numbers of TF binding sites and the PIC, flanked by BRG1-bound nucleosomes. Although examples of association between BRG1 and TFs have been previously described (*Trotter and Archer, 2008*; *Reisman et al., 2009*; *Euskirchen et al., 2011*), this specific configuration of BRG1 binding to nucleosomes flanking the TF binding sites has not been generally recognised. In yeast, it has however been reported that TF binding sites are rather flanked by nucleosomes bound by ISWI and CHD remodellers (*Zentner et al., 2013*).

While there have been many examples of co-localization between BRG1 and TFs, this is the first description of active genome-wide BRG1 recruitment. siRNA-mediated MITF or SOX10 silencing

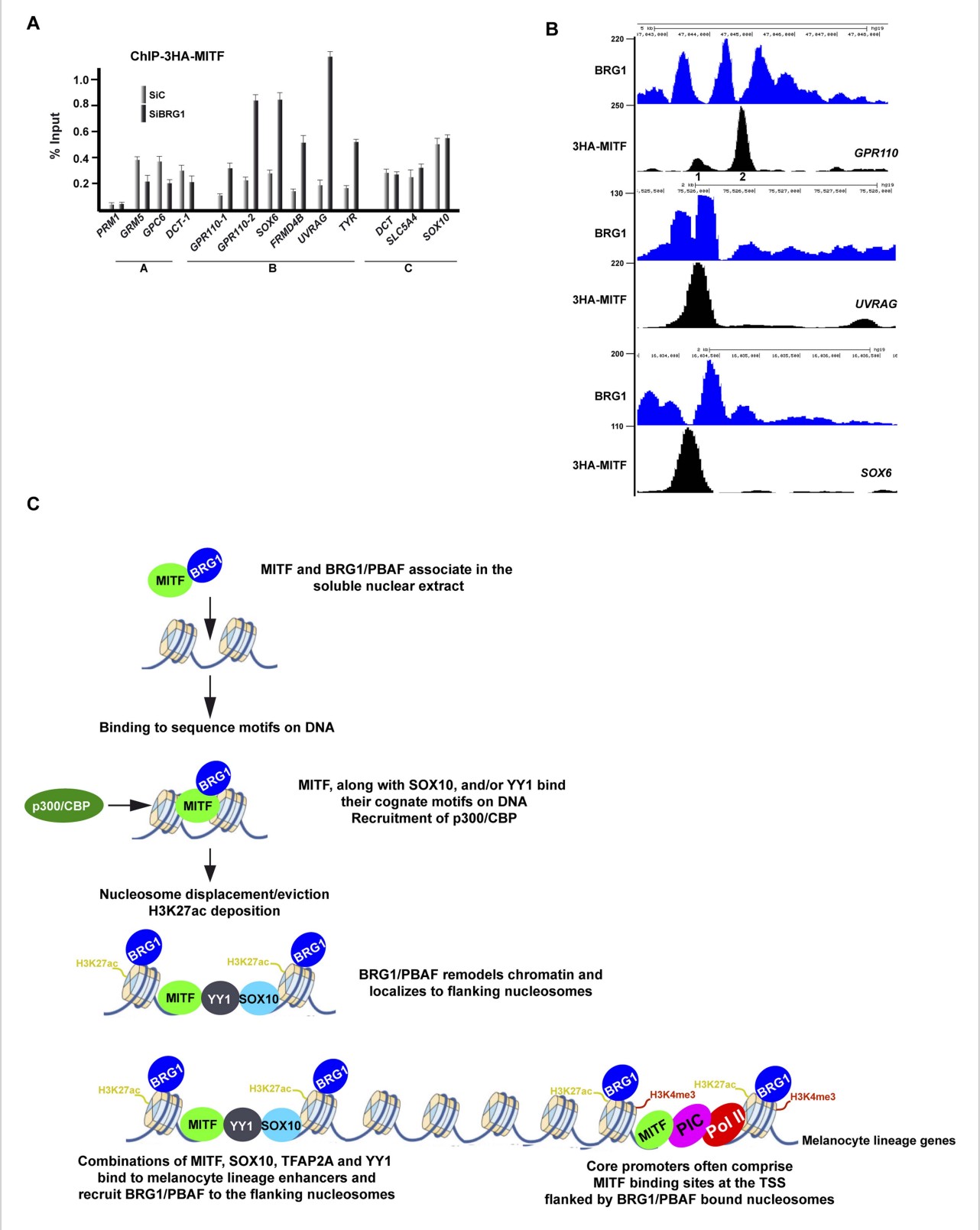

**Figure 9**. BRG1 controls dynamics of MITF binding. (**A**) ChIP-qPCR of 3HA-MITF in 501Mel-CL8 cells at the indicated loci following transfection with siLuc or siBRG1. The protamine 1 locus (PRM1) was used as a negative control. (**B**) UCSC screenshots illustrating binding of MITF between two BRG1-occupied nucleosomes at selected loci assayed by ChIP-qPCR in panel **A**. sThe GPR110-1 and GPR110-2 sites assayed in Panel **A** are indicated in panel **B**.

*Figure 9. continued on next page*

*Figure 9. Continued*

(**C**) A model for regulatory elements in the melanocyte lineage. Melanocyte lineage enhancers comprise combinations of MITF, SOX10, YY1, and also TFAP2A and ETS1 (not represented for simplicity. Note also that Pol II and the PIC are present at active enhancers where enhancer RNAs are made. For simplicity these are also not represented.) bound to a nucleosome-depleted region. MITF but also these other factors recruit BRG1/PBAF to the nucleosomes flanking the combinations of transcription factors. BRG1/PBAF also occupies the nucleosomes flanking the TSS and a subset of these promoters further comprises a MITF binding site close to the TSS.

identified sites to which BRG1 was recruited either individually or cooperatively by these TFs. Moreover, we show that MITF associates with PBAF in the soluble nuclear fraction suggesting that they form a complex in the nucleoplasm and are recruited simultaneously to the chromatin to create the NDRs (*Figure 9C*).

Chromatin remodelling has been shown to facilitate TF binding, a good example being that of TAL1 where BRG1 repositions nucleosomes flanking its binding sites (*Hu et al., 2011*). In contrast, we observed that MITF occupancy of many co-occupied sites either shows little change or is increased following siBRG1 silencing suggesting that BRG1 regulates the dynamics of MITF binding. TF binding to chromatin is often extremely dynamic with ChIP capturing only a snapshot of their occupancy (*Voss and Hager, 2014*). The increased MITF occupancy seen in ChIP upon BRG1 silencing suggests an increase in its time of residence at occupied sites.

Several mechanisms may explain the increased MITF binding upon BRG1 loss. The human genome comprises many more consensus E-box elements than are occupied by MITF. The excess of consensus as well as degenerate E-boxes could potentially act as a sink thus limiting the pool of MITF for binding to functional sites. One function of BRG1-driven dynamics may therefore be to limit MITF sequestration at non-functional sites and ensure a pool of MITF for binding to functional sites. Alternatively, BRG1 may be required to establish the NDRs for MITF binding, for example after mitosis, or at specific stages of the cell cycle. As siBRG1 cells are post-mitotic and senescent, the increased MITF occupancy may reflect a new steady state where MITF remains more stably bound to the NDRs established while BRG1 was still present compared to cycling cells where the NDRs are established and erased in a more dynamic manner. Irrespective of the underlying causes, our results indicate that BRG1 regulates dynamic MITF interactions with chromatin.

## Materials and methods

### Generation of 501Mel cells stably expressing F-H-MITF

501Mel cells cultured in RPMI 1640 medium (Sigma, St Louis, MO, USA) supplemented with 10% fetal calf serum (FCS) were transfected with a CMV-based vector expressing Flag-HA-tagged MITF and a vector encoding puromycin resistance. Transfected cells were selected with puromycin (3 µg/ml), and the expression of MITF verified by western blot using the MITF antibody ab-1 (C5) from Neomarkers, the 12CA5 HA antibody (Roche, Basel, Switzerland), or the M2 Flag antibody (Sigma). Details of other cell culture are provided in Supplementary information. Hermes 3A cells were obtained from the University of London St Georges repository.

### Tandem immunoaffinity purification and mass-spectrometry

Cell extracts were prepared essentially as previously described and subjected to tandem Flag-HA immunoprecipitation (*Drané et al., 2010*). Cells were lysed in hypotonic buffer (10 mM Tris–HCl at pH 7.65, 1.5 mM MgCl$_2$, 10 mM KCl) and disrupted by Dounce homogenizer. The cytosolic fraction was separated from the pellet by centrifugation at 4˚C. The nuclear soluble fraction was obtained by incubation of the pellet in high salt buffer (final NaCl concentration of 300 mM) and then separated by centrifugation at 4˚C. To obtain the nuclear insoluble fraction (chromatin fraction), the remaining pellet was digested with micrococcal nuclease and sonicated. Tagged proteins were immunoprecipitated with anti-Flag M2-agarose (Sigma), eluted with Flag peptide (0.5 mg/ml), further affinity purified with anti-HA antibody-conjugated agarose (Sigma), and eluted with HA peptide (1 mg/ml). The HA and Flag peptides were first buffered with 50 mM Tris–HCl (pH 8.5), then diluted to 4 mg/ml in TGEN 150 buffer (20 mM Tris at pH 7.65, 150 mM NaCl, 3 mM MgCl$_2$, 0.1 mM EDTA, 10% glycerol,

0.01% NP40), and stored at −20°C until use. Between each step, beads were washed in TGEN 150 buffer. Complexes were resolved by SDS-PAGE and stained using the Silver Quest kit (Invitrogen, La Jolla, CA, USA). Mass-spectrometry was performed at the Taplin Biological Mass Spectrometry Facility (Harvard Medical School, Boston, MA).

## Cell culture and lentiviral infections

Melanoma cell lines SK-Mel-28 and 501Mel were grown in RPMI 1640 medium (Sigma) supplemented with 10% FCS. 293T cells were grown in Dulbecco's modified Eagle's medium supplemented with 10% FCS and penicillin/streptomycin (7.5 µg/ml). Hermes-3A cells were grown in RPMI 1640 medium (Sigma) supplemented with 10% FCS, 200 nM TPA, 200 pM cholera toxin, 10 ng/ml human stem cell factor (Invitrogen), 10 nM endothelin-1 (Bachem, Bubendorf, Switzerland), and penicillin/streptomycin (7.5 µg/ml). All lentiviral shRNA vectors were obtained from Sigma (Mission sh-RNA series) in the PLK0 vector. The following constructs were used. shBRG1 (TRCN0000015549) and shMITF (TRCN0000019119). In each case between $5 \times 10^5$ and $1 \times 10^6$ cells were infected with the indicated shRNA lentivirus vectors and all experiments were performed at least in triplicate. siRNA knockdowns were performed with the corresponding ON-TARGET-plus SMARTpools purchased from Dharmacon Inc. (Chicago, Il., USA). Control siRNA directed against luciferase was obtained from Eurogentec (Seraing, Belgium). siRNAs were transfected using Lipofectamine RNAiMax (Invitrogen).

## Transfections, extract preparation and antibodies

Transient and stable transfections were performed with 5 µg of expression vectors and using FuGENE 6 reagent (Roche) following the manufacturer's instructions. Medium was replaced 24 hr and cells were collected 48 hr after transfection. Cells lysis was performed using LSDB 500 buffer (500 mM KCl, 25 mM Tris at pH 7.9, 10% glycerol, 0.05% NP-40, 1 mM DTT, and protease inhibitor cocktail). Up to 3 mg of whole cell extracts were diluted in LDSB without KCl to obtain a final concentration of 100 mM KCl and incubated for 12 hr with 5 µg of specific antibody and 50 µl Slurry of protein-G sepharose (GE Healthcare). Beads were washed 3 times in LSDB 300, twice in LSDB 150, and boiled in Laemmli buffer before protein separation by SDS–PAGE. For flag immunoprecipitations, extracts were incubated with 50 µl Slurry of Anti-Flag M2-agarose affinity gel (Sigma) and washed similarly prior to elution with Flag peptide (0.5 mg/ml). Immunoblots were performed with the following antibodies: MITF (MS-771-P; Interchim), BRG1 (ab110641; Abcam, Cambridge, UK), HERC2 (612366; BD Transduction Laboratories, Sparks, MD), USP11 (3263-1; Epitomics, Burlingame, CA), USP7 (#4833; Cell Signaling, Danvers, MA), TRRAP (2TRR-2D5; IGBMC), NEURL4 (sc-243603; scbt, Santa Cruz, CA), actin (2D7; IGBMC), XRCC6 (sc-17789; scbt), XRCC5 (sc-5280; scbt), BAF170 (A301-038A; Bethyl Laboratories, Montgomery, TX), BAF155 (sc-10756; scbt), BAF250A (sc-373784; scbt), BAF250B (sc-32762; scbt), BAF200 (ab56082; Abcam), BAF53A (ab131272; Abcam), CHD7 (ab31824; Abcam), BAF180 (ab137661; Abcam), BAF60A (#611728; BD Transduction Laboratories), BAF60B (ab166622; Abcam), SOX10 (ab155279; Abcam), CREB (#06-863; Upstate Millipore, Molsheim, France).

## Mice and genotyping

The *Smarca4*[lox/lox] and *Tyr::Cre* strains have been described previously (*Indra et al., 2005*; *Delmas et al., 2003*). Genotyping of F1 offspring was carried out by PCR analysis of genomic tail DNA with primers detailed in the respective publications. All animals were handled according to institutional and national guidelines and policies.

## Chromatin immunoprecipitation and sequencing

BRG1 ChIP experiments were performed on native Mnase-digested chromatin. $5 \times 10^7$ to $5 \times 10^8$ freshly harvested 501Mel cells were resuspended in 2 ml ice-cold hypotonic buffer (0.3M Sucrose, 60 mM KCl, 15 mM NaCl, 5 mM MgCl$_2$, 0.1 mM EDTA, 15 mM Tris–HCl [pH 7.5], 0.5 mM DTT, 0.1 mM PMSF, protease inhibitor cocktail) and cytoplasmic fraction was released by incubation with 2 ml of lysis-buffer (0.3M sucrose, 60 mM KCl, 15 mM NaCl, 5 mM MgCl$_2$, 0.1 mM EDTA, 15 mM Tris–HCl [pH 7.5], 0.5 mM DTT, 0.1 mM PMSF, PIC, 0.5% (vol/vol) IGEPAL CA-630) for 10 min on ice. The suspension was layered onto a sucrose cushion (1.2 M sucrose, 60 mM KCl, 15 mM NaCl, 5 mM MgCl$_2$, 0.1 mM EDTA, 15 mM Tris–HCl [pH 7.5], 0.5 mM DTT, 0.1 mM PMSF, PIC) and centrifuged for 25 min at 4700 rpm in a swing rotor. The nuclear pellet was resuspended in digestion buffer (0.32M

sucrose, 50 mM Tris–HCl [pH 7.5], 4 mM MgCl₂, 1 mM CaCl₂, 0.1 mM PMSF) and subjected to Micrococcal Nuclease digestion for 5 min at 37°C. The reaction was stopped by addition of EDTA and suspension chilled on ice for 10 min. The suspension was cleared by centrifugation at 10,000 rpm (4°C) for 10 min and supernatant (chromatin) was used for further purposes. Chromatin was digested to around 80% of mono-nucleosomes as judged by extraction of the DNA and agarose gel electrophoresis. SOX10 and 3HA-MITF ChIP experiments were performed on 0.4% PFA-fixed chromatin isolated from 501Mel and Cl8 cells, respectively according to standard protocols as previously described (*Strub et al., 2011*). ChIP-seq libraries were prepared as previously described and sequenced on the Illumina Hi- seq2500 as single-end 50-base reads (*Herquel et al., 2013*). After sequencing, peak detection was performed using the MACS software ([*Zhang et al., 2008*] http://liulab.dfci.harvard.edu/MACS/). Peaks were then annotated with GPAT (*Krebs et al., 2008*) using a window of ±10 kb (or as indicated in the figures) relative to the transcription start site of RefSeq transcripts. Global clustering analysis and quantitative comparisons were performed using seqMINER ([*Ye et al., 2011*] http://bips.u-strasbg.fr/seqminer/) and R (http://www.r-project.org/). The public human foreskin melanocyte H3K27ac data were taken from the Geo entry GSM958157.

De novo motif discovery on FASTA sequences corresponding to windowed peaks was performed using MEME-ChIP. Motif correlation matrix was calculated with in-house algorithms using JASPAR database.

## mRNA preparation, quantitative PCR and RNA-seq

mRNA isolation was performed according to standard procedure (Qiagen kit, Venlo, Holland). qRT-PCR was carried out with SYBR Green I (Qiagen) and Multiscribe Reverse Transcriptase (Invitrogen) and monitored by a LightCycler 480 (Roche). Detection of Actin gene was used to normalize the results. RNA-seq was performed essentially as previously described (*Herquel et al., 2013*). Gene ontology analyses were performed using the functional annotation clustering function of DAVID (http://david.abcc.ncifcrf.gov/). Primers for RT-qPCR and ChIP-qPCR were designed using Primer 3 and are listed in *Supplementary file 5*.

## Motif analysis

Searching of known TF motifs from the Jaspar 2014 motif database at BRG1-bound sites was made using FIMO (*Grant et al., 2011*) within regions of 200 bp around peak summits, FIMO results were then processed by a custom Perl script which computed the frequency of occurrence of each motif. To assess the enrichment of motifs within the regions of interest, the same analysis was done 100 times on randomly selected regions of the same length as the BRG1 bound regions and the results used to compute an expected distribution of motif occurrence. The significance of the motif occurrence at the BRG1-occupied regions was estimated through the computation of a Z-score (z) with $z = (x − μ)/σ$, where: − x is the observed value (number of motif occurrence), − μ is the mean of the number of occurrences (computed on randomly selected data), − σ is the standard deviation of the number of occurrences of motifs (computed on randomly selected data). The source code is accessible at https://github.com/slegras/motif-search-significance.git.

## Immunostaining

Biopsies of dorsal skin were isolated and fixed overnight in 4% paraformaldehyde, washed with PBS, dehydrated, paraffin embedded, and sectioned at 5 μm. For antigen retrieval, the sections were incubated with 10 mM sodium citrate buffer, within a closed plastic container placed in a boiling waterbath, for 20 min. Sections were permeabilised with 3 × 5 min 0.1% Triton in PBS, blocked for 1 hr in 5% skim milk in PBS, and incubated overnight in 5% milk with primary antibodies. The following antibodies were used: goat anti-Dct at dilution of 1/1000 (Santa Cruz Biotechnology, sc-10451) and rabbit anti-Sox10, at 1/2000 (Abcam, ab155279). Sections were washed 3 × 5 min 0.1% Triton in PBS, and incubated with secondary antibodies, Alexa 488 donkey-anti-goat, and Alexa 555 donkey-anti-rabbit (Invitrogen) for 2 hr. Sections were subsequently incubated with 1/2000 Hoechst nuclear stain for 10 min, washed 3 × 5 min in PBS, dried and mounted with Vectashild.

## Senescence-associated β-galactosidase assay

The senescence-associated β-galactosidase staining kit from Cell signaling technology (Beverly, MA, USA) was used according to the manufacturer's instructions to histochemically detect β-galactosidase activity at pH 6.

## Acknowledgements

We thank, S Gygi and R Tomaino for mass-spectrometry analysis, A Hamiche for advice on tandem immunopurification, L Larue for the *Tyr*-Cre mice, P Chambon and D Metzger for the floxed *Smarca4* mice, E Sviderskaya and D Bennet for the Hermes cells, B Jost, and all the staff of the IGBMC high throughput sequencing facility, a member of 'France Génomique' consortium (ANR10-INBS-09-08). This work was supported by grants from the CNRS, the INSERM, the Fondation ARC pour la Recherche contre le Cancer, the Ligue Nationale et Départementale Région Alsace contre le Cancer and the Institut National du Cancer (INCa) 2011-1-PL BIO-03-INSERM 16-1, the ANR-10-LABX-0030-INRT French state fund through the Agence Nationale de la Recherche under the frame programme Investissements d'Avenir labelled ANR-10-IDEX-0002-02. ID is an 'équipe labellisée' of the Ligue Nationale contre le Cancer. The data in this paper have been submitted to the Geo database under the reference GSE61967.

## Additional information

### Competing interests

ID: Reviewing editor, *eLife.* The other authors declare that no competing interests exist.

### Funding

| Funder | Grant reference | Author |
| --- | --- | --- |
| Ligue Contre le Cancer | équipe labellisée | Irwin Davidson |
| Centre National de la Recherche Scientifique (National Center for Scientific Research) | | Irwin Davidson |
| Institut national de la santé et de la recherche médicale (National Institute of Health and Medical Research) | | Irwin Davidson |
| Fondation ARC pour la Recherche sur le Cancer (ARC Foundation for Cancer Research) | | Irwin Davidson |
| Institut National du Cancer | | Irwin Davidson |
| Agence Nationale de la Recherche (L' Agence Nationale de la Recherche) | ANR-10-LABX-0030-INRT | Irwin Davidson |
| National Institutes of Health (NIH) | AR062547 | Robert A Cornell |

The funders had no role in study design, data collection and interpretation, or the decision to submit the work for publication.

### Author contributions

PL, TS, GM, Conception and design, Acquisition of data, Analysis and interpretation of data, Drafting or revising the article; DK, Acquisition of data, Analysis and interpretation of data; CK, SLG, ID, Conception and design, Analysis and interpretation of data, Drafting or revising the article; HS, EVO, Contributed to generation and analysis of the TFAP2A ChIP-seq data used in this study, Acquisition of data, Analysis and interpretation of data, Contributed unpublished essential data or reagents; HI, Contributed to the acquisition and analysis of the H3K27ac ChIP-seq data from primary proliferative melanoma used in this study, Acquisition of data, Analysis and interpretation of data, Contributed unpublished essential data or reagents; RS, Contributed to the acquisition and analysis of the MITF ChIP-seq data used in this study, Acquisition of data, Analysis and interpretation of data; SA, Contributed to the acquisition and analysis of the H3K27ac ChIP-seq data from primary proliferative melanoma used in this study, Acquisition of data, Analysis and interpretation of data, Drafting or revising the article; RAC, Contributed to the acquisition and analysis of the TFAP2A ChIP-seq data used in this study, Acquisition of data, Analysis and interpretation of data, Drafting or revising the article

### Ethics

Animal experimentation: Animal experiments were performed in compliance with National Animal Care Guidelines (European Commission directive 86/609/CEE; French decree no. 87-848).

## Additional files

### Supplementary files

• Supplementary file 1. Related to *Figure 1*. Mass spectrometry identification of MITF partners. Excel table showing the data from the mass spectrometry analysis. Page 1 shows a summary of the number of peptides and proteins identified in the two experiments in the soluble nuclear and chromatin associated fractions. Page 2 lists the proteins identified uniquely in the F-H-MITF immunoprecipitates from the soluble nuclear fraction along with the number of peptides for each protein. Page 3 lists the proteins identified uniquely in the F-H-MITF immunoprecipitates from the chromatin-associated fraction along with the number of peptides for each protein.

• Supplementary file 2. Related to *Figure 2*. Excel spread sheet of genes specifically and commonly regulated by BRG1 and MITF knockdown in 501Mel and Hermes 3A cells along with the appropriate gene ontology, see Figures S3B–D.

• Supplementary file 3. Excel spread sheet of genes with associated BRG1 occupancy (either ±10 kb, or ±30 kb with respect to TSS) and regulated in shBRG1 along with the appropriate gene ontology as described in *Figure S5E*.

• Supplementary file 4. Excel spread sheet of genes associated with BRG1 and MITF co-occupied sites or MARES along with their gene ontology.

• Supplementary file 5. Excel spread sheet of primer sequences used for RT-qPCR and ChIP-qPCR.

### Major datasets

The following dataset was generated:

| Author(s) | Year | Dataset title | Dataset ID and/or URL | Database, license, and accessibility information |
|---|---|---|---|---|
| Laurette P, Strub T, Koludrovic D, Keime C, Le Gras S, Seberg H, Van Otterloo E, Imrichova H, Siddaway R, Aerts S, Cornell RA, Mengus G, Davidson I | 2015 | BRG1 recruitment by transcription factors MITF and SOX10 defines a specific configuration of regulatory elements in the melanocyte lineage | http://www.ncbi.nlm.nih. gov/geo/query/acc.cgi? acc=GSE61967 | Publicly available at the NCBI Gene Expression Omnibus: GSE61967. |

The following previously published dataset was used:

| Author(s) | Year | Dataset title | Dataset ID and/or URL | Database, license, and accessibility information |
|---|---|---|---|---|
| UCSF-UBC CENTER | 2012 | H3K27ac ChIP-Seq analysis of Penis, Foreskin, Melanocyte skin03; A15584-1 | http://www.ncbi.nlm.nih. gov/geo/query/acc.cgi? acc=GSM958157 | Publicly available at the NCBI Gene Expression Omnibus: GSM958157. |

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
