## [Decision Letter]

Thank you for sending your work entitled “Combinatorial binding of MITF and BRG1 define specific chromatin configurations of melanoma cell regulatory elements” for consideration at *eLife*. Your article has been favorably evaluated by James Manley (Senior editor), Michael Green (Reviewing editor), and two expert reviewers (Collin Goding and Narendra Wajapeyee).

The Reviewing editor and the reviewers discussed their comments before we reached this decision, and the Reviewing editor has assembled the following comments to help you prepare a revised submission.

The paper from Laurette et al. presents the first comprehensive Mass Spec analysis of MITF-interacting factors, and then goes on focus on the role of BRG1 containing-chromatin-remodelling complex in melanocytes and melanoma. They undertake genome-wide ChIP-seq analsysis of BRG1, and tie this to RNA-seq following BRG1 knock-down. This is then intersected with previously published MITF ChIP-seq and RNA seq-data sets, as well a new SOX10 ChIP seq analysis. Further intersection is performed using a previously published YY1 data set as well as ENCODE H3K27 ChIP-seq. A melanocyte-specific BRG1 knockout mouse is also presented for the first time.

Some results are not new and the novel insights into the biology of melanocytes and melanoma are limited. As readily acknowledged by the authors of this manuscript, previous work from the de la Serna lab has established that the SWI/SNF complex is a co-factor for MITF, while the Vachtenheim lab has shown that the SWI/SNF complex is required in cell lines for MITF expression. As such, it is not surprising that either knockout of BRG1 in melanocytes gives pigmentation phenotype, or that many MITF targets are also BRG1 targets.

That said, the data presented here, by adopting a genome-wide approach go much further than previous work, and while some of the genome-wide analysis could be classified as somewhat descriptive (as is the nature of most genome-wide ChIP-/RNA-seq analyses), the results presented will provide an invaluable resource for the melanocyte-/melanoma field.

For example, previous work could not distinguish between a requirement of the SWI/SNF complex in regulating MITF target genes because it is also needed for MITF expression or because it acts as a co-factor for MITF at those targets. In this paper the genome-wide approaches do nicely distinguish between those possibilities, and moreover, go on to show SOX10 genome-wide occupancy and co-occupancy with the SWI/SNF complex. The genome-wide analyses are very nicely done and provide key insights into the regulatory landscape of melanocytes and melanoma.

The Mass Spec analysis that identifies a wide range of MITF interacting factors is also novel and will again provide an excellent resource for the field.

Because of this, the positive aspects of the work presented greatly outweigh the lack of novelty in a few areas.

Specific comments:

1) In the subsection headed “BRG1 regulates an extensive gene expression programme essential for proliferation of 156 melanoma cells in vitro”, the authors state that 'MITF and BRG1 cooperate to activate these critical pathways', yet at this stage of the paper the authors cannot state whether the effects arise because of BRG1-mediated regulation of MITF or via BRG1 acting as an MITF co-factor. The word 'cooperate' to my mind tends to indicate the latter, rather than the former. While this issue is resolved later, at this stage it is not clear which of the two possibilities is more important.

2) In the subsection headed “BRG1 and MITF regulate gene expression in human melanocytes”, the authors mention that in Hermes cells BRG1 silencing affected fewer genes than in melanoma cells. Is this because fewer genes are expressed or highly expressed? Is BRG1 protein level greatly reduced? Some comment to discuss why there is this difference would be useful.

3) The melanocyte-specific BRG1 knock out mice clearly show a pigmentation phenotype. Almost certainly this arises because of the requirement for BRG1 in MITF expression and a loss of melanoblasts during development. However, formally the authors have not distinguished between a shut-down of pigmentation genes vs. loss of melanoblasts in development.

4) In Figure 3, a higher magnification image of the senescent cells, perhaps as an additional panel, would be useful.

5) Some parts of the Results section of the manuscript read like discussion and are speculative. For example, the first subsection of the Results discusses the possible role of the MITF interacting protein without providing any experimental evidence. This should therefore either be moved to the Discussion section or completely removed.

6) The effect of BRG1 knockdown on MITF expression is interesting. In particular because authors see downregulation of MITF due to BRG1 knockdown in melanoma cells but not in melanocytes. What is the reason behind this differential effect of BRG1 knockdown on MITF expression and what are its implications for regulation of MITF mediated transcription regulation by BRG1?

7) BRG1 has limited impact on the gene transcription in melanocytes. Therefore, it is possible that the observed effect of BRG1 on melanocytes may be due to transcription regulation independent function of BRG1. It will be important to rule out this possibility.

8) A previous study shows that BRG1 can function as a pro-survival gene independent of MITF (Ondrusova et al., PloS One, 2013). How do the authors reconcile their results in light of these studies?

---

## [Author Response]

*1) In the subsection headed “BRG1 regulates an extensive gene expression programme essential for proliferation of 156 melanoma cells in vitro”, the authors state that 'MITF and BRG1 cooperate to activate these critical pathways', yet at this stage of the paper the authors cannot state whether the effects arise because of BRG1-mediated regulation of MITF or via BRG1 acting as an MITF co-factor. The word 'cooperate' to my mind tends to indicate the latter, rather than the former. While this issue is resolved later, at this stage it is not clear which of the two possibilities is more important*.

We agree with the referees: we have modified this sentence to remove the reference to cooperation.

*2) In the subsection headed “BRG1 and MITF regulate gene expression in human melanocytes”, the authors mention that in Hermes cells BRG1 silencing affected fewer genes than in melanoma cells. Is this because fewer genes are expressed or highly expressed? Is BRG1 protein level greatly reduced? Some comment to discuss why there is this difference would be useful*.

We have compared the gene expression profiles of 501Mel and Hermes 3A cells by calculating the number of genes with increasing RPKM values as a % of the total expressed genes. As can be seen from Figure 10, the distribution of gene expression is very similar in both cell types. Thus, there are not fewer highly expressed genes in Hermes cells than in 501Mel cells.

Author response image 1.**DOI:**
http://dx.doi.org/10.7554/eLife.06857.026

Also, the immunoblot analyses in Figure 3 shows that Brg1 protein levels are comparably repressed in Hermes 3A and 501Mel cells and importantly that the knockdown in Hermes cells is sufficient to induce senescence and eventual cell death. Thus the difference in the number of affected genes is unlikely to be explained by inefficient BRG1 silencing in Hermes cells. Our results rather suggest that the function of PBAF with BRG1 while critical is less ubiquitous in these cells than in 501Mel. Perhaps other variants of the SWI/SNF complexes with the related BRM subunit also play an important role in gene regulation in these cells. Our RNA-seq data shows that BRM is expressed in Hermes cells and its expression (as well as the genes encoding all other SWI/SNF subunits) is not affected by BRG1 silencing. There is therefore a potential for redundancy in these cells between BRG1 and BRM, although it is important to note that this is not the case in 501Mel cells where BRM is also expressed. Several of the additional comments below are related to this point. In the future it will be important to address this issue, but it is beyond the scope of the present study. We have modified the Discussion to take this issue into account.

*3) The melanocyte-specific BRG1 knock out mice clearly show a pigmentation phenotype. Almost certainly this arises because of the requirement for BRG1 in MITF expression and a loss of melanoblasts during development. However, formally the authors have not distinguished between a shut-down of pigmentation genes vs. loss of melanoblasts in development*.

We agree with the referees on this point and this is why we came back to this point in the Discussion to say that ‘while it is formally possible that melanoblasts develop in these mice, but do not produce melanin/melanosomes for pigmentation, the arrested proliferation and cell death seen in melanoma cells and melanocytes in vitro, rather indicate that loss of pigmentation resulted from absence of melanocytes in these animals*’*. Nevertheless, we have added new data to Figure 4 showing immunostaining of hair shafts from *Smarca4*^lox/+^ or *Smarca4*^mel-/-^ animals with antibodies against Dct and Sox10. In *Smarca4*^mel+/-^ mice, Sox10 stains the nucleus of mature bulb melanocytes and Dct shows cytoplasmic staining. In contrast, there is no staining with these antibodies in *Smarca4*^mel-/-^ animals. In Hermes melanocytes in vitro*, DCT* expression is strongly repressed by shBRG1 thus one could argue that in vivo the absence of Dct staining does not exclude the possible presence of immature or partially differentiated melanocytes that do not express this marker. However *SOX10* expression is not affected by BRG1 silencing in Hermes melanocytes in vitro, and is a marker for immature transient amplifying cells, yet there is no Sox10 expression in the mutant hair shafts in vivo. There are thus no identifiable melanocytes in the *Smarca4*^mel-/-^ hair follicles. We have added this data to the revised version along with a sentence in the Discussion.

*4) In*
Figure 3*, a higher magnification image of the senescent cells, perhaps as an additional panel, would be useful*.

As requested by the referees, we have modified this figure to show a higher magnification.

*5) Some parts of the Results section of the manuscript read like discussion and are speculative. For example, the first subsection of the Results discusses the possible role of the MITF interacting protein without providing any experimental evidence. This should therefore either be moved to the Discussion section or completely removed*.

We agree with the comment of the referees and we have moved parts of this section to the Discussion.

*6) The effect of BRG1 knockdown on MITF expression is interesting. In particular because authors see downregulation of MITF due to BRG1 knockdown in melanoma cells but not in melanocytes. What is the reason behind this differential effect of BRG1 knockdown on MITF expression and what are its implications for regulation of MITF mediated transcription regulation by BRG1*?

In 501Mel cells, our results suggest two important contributions of BRG1 to MITF expression. The first is via SOX10. We showed the BRG1 silencing strongly down-regulates SOX10 expression and that SOX10 is required to activate MITF expression. The second pathway by which BRG1 may act to regulate MITF expression is through MITF auto-regulation. This is suggested by the colocalisation of MITF and BRG1 at H3K27ac marked elements of the MITF locus suggesting that these may be enhancer elements via which MITF may regulate its own expression. Clearly the situation is very different in melanocytes. In melanocytes, SOX10 expression is not reduced by BRG1 silencing and thus may contribute to maintenance of MITF expression. A role for SOX10 in modulating MITF expression in Hermes 3A cells has previously been described by the Ronai lab (Shah et al., 2010). In melanocytes in which BRG1 is silenced, MITF remains expressed, but 30% of the genes regulated by MITF are down-regulated. These data show that part of the MITF-directed gene expression program in melanocytes is co-regulated by BRG1.

*7) BRG1 has limited impact on the gene transcription in melanocytes. Therefore, it is possible that the observed effect of BRG1 on melanocytes may be due to transcription regulation independent function of BRG1. It will be important to rule out this possibility*.

We agree with the referees that there may be transcription independent functions of BRG1 such as in DNA and repair and recombination as already documented by other studies. Nevertheless, there is an effect of BRG1 silencing on transcription in melanocytes and while it is less extensive than in 501Mel cells, it is sufficient to induce their senescence. It is not because the number of genes that are BRG1 dependent is reduced that BRG1 has no role in transcription in these cells.

*8) A previous study shows that BRG1 can function as a pro-survival gene independent of MITF (Ondrusova et al., PloS One, 2013). How do the authors reconcile their results in light of these studies*?

Our data show that BRG1 is recruited to regulatory elements not only by MITF, but also by SOX10. Moreover, the binding motifs for a variety of factors are enriched at BRG1-occupied sites suggesting that it can be recruited to the genome of melanoma cells by many other transcription factors. This is also in accordance with the fact that BRG1 silencing affects expression of many more genes than MITF. BRG1 is likely therefore to act as a cofactor for many other transcription factors and thus explaining how it can act independently of MITF. A sentence on this has been added to the Discussion.

Also Dr. Peter Rodgers, the Features editor, suggested that we modify the title to read: ‘Combinatorial binding of the transcription factor MITF and the transcription activator BRG1 define specific chromatin configurations of melanoma cell regulatory elements’. We have chosen: ‘Transcription factor MITF and the remodeller BRG1 define chromatin organisation at regulatory elements in melanoma cells’.